# Phenological Adaptation of Wheat Varieties to Rising Temperatures: Implications for Yield Components and Grain Quality

**DOI:** 10.3390/plants13202929

**Published:** 2024-10-18

**Authors:** Davide Gulino, Marta S. Lopes

**Affiliations:** Sustainable Field Crops Program, Institute of Agrifood Research and Technology (IRTA), 25198 Lleida, Spain; davide.gulino@irta.cat

**Keywords:** heat, drought, wheat grain quality, protein, late sowing

## Abstract

This study examined the effects of late sowing, water restrictions, and interannual weather variations on wheat grain yield and quality through field trials in Spain over two growing seasons. Delayed sowing and water scarcity significantly reduced yields, with grain quality mainly affected under rainfed conditions. Early-maturing varieties performed better in these conditions, benefiting from lower temperatures and extended grain-filling periods, leading to higher solar radiation interception, potentially increased photosynthetic activity, and improved yields. These varieties also saved water through reduced total cumulative evapotranspiration from sowing to maturity (ETo TOT), which was advantageous in water-limited environments. In contrast, late-maturing varieties were exposed to higher maximum temperatures during grain filling and experienced greater ETo TOT, leading to lower yields, reduced hectoliter weight, and a lower P/L ratio (tenacity/extensibility). This study highlighted the importance of optimizing temperature exposure and evapotranspiration for improved grain yield and quality, especially under climate change conditions with higher temperatures and water shortages. Notably, it established, for the first time, the importance of phenology on wheat quality of different varieties, suggesting that targeted selection for specific phenology could mitigate the negative impacts of heat stress not only on grain yield but also on grain quality.

## 1. Introduction

Rising temperatures have contributed to stagnation in wheat yields in many regions across Europe [1,2] and on a global scale [3,4]. Models predict even more severe impacts in the coming years [5,6]. In this context, the Mediterranean basin has been identified as one of the major climate change hotspots due to significant variations in temperature and precipitation [7]. Specifically in Spain, several studies on winter wheat have projected yield declines due to increased temperatures [8] and disruptions in vernalization requirements in the northern regions of the country [9]. Other studies have indicated that the expected yield reductions are likely to be more severe in southern Spain compared to the northern regions [10,11]. Rising temperatures associated with late sowing were highlighted as a key factor in shortening the growing period, thereby contributing to reduced yields [5,12,13,14,15,16,17]. With respect to grain quality, studies have shown that grain protein concentration increased under drought stress and higher temperatures due to reduced starch accumulation [18], reinforcing the general principle that grain yield and protein concentration are negatively correlated. Several authors [18,19,20,21] have explored how temperature can alter gluten composition in wheat and its subsequent impact on dough strength, emphasizing that variations in baking properties cannot be explained solely by differences in protein concentration [22]. Research has identified a specific temperature threshold (around 30 °C) beyond which the positive relationship between grain protein concentration and dough strength does not hold, with the threshold being even lower for heat-sensitive varieties. At higher temperatures, this correlation can even turn negative due to the differential effects of heat on protein fractions: gliadin accumulation is less affected than glutenin, leading to an increased gliadin/glutenin ratio [23,24]. Additionally, water restrictions can exacerbate the effects of heat stress, further limiting the crop’s capacity to develop properly and achieve optimal yields and quality [25]. 

Given the negative impacts of temperature and drought stress on wheat described above, it is essential to identify adaptive strategies to mitigate the effects of late sowing and water stress to sustain wheat productivity under changing climatic conditions [17]. One of the most used strategies is the deployment of wheat varieties adapted to these challenging conditions. Based on simulations, Liu et al. [16] tested various genotypes in a rice–wheat double cropping system and found that those with higher radiation use efficiency, extended juvenile phases, and higher grain filling rates were more effective in mitigating yield losses due to delayed sowing and rising temperatures. Other studies have indicated that under late sowing, genotypes with shorter vegetative periods and longer reproductive phases performed well [26]. Moreover, key parameters for selecting high-yield genotypes were days to anthesis and days to maturity, with faster anthesis and slower maturity correlating with higher yields [26]. Giannakopoulos et al. [27] reported that using longer growing cycle cultivars, as well as implementing early sowing dates, were effective strategies to counteract the shortening of the growing season due to climate change. Asseng et al. [28] suggested an adapted trait combination consisting of longer growing cycles paired with cultivars that have higher grain filling rates, which could potentially increase yields on a global scale. Ruiz-Ramos et al. [11] conducted simulations in the region where this study was carried out (near Lleida, northeastern Spain) and explored adaptations such as removing vernalization requirements, implementing supplementary irrigation, and adopting earlier sowing dates. These variations in results are largely influenced by specific environmental factors, such as precipitation distribution patterns and the timing of heat stress, indicating that both strategies can be effective depending on the conditions. However, in some situations, planting earlier to extend the crop cycle, as suggested by some authors [11], may not be feasible, such as in double cropping systems. This supports the importance of developing varieties adapted to high temperatures.

Despite the body of work already available, there remains a lack of comprehensive research on how these genotypic adaptation strategies affect not just yield but simultaneously key quality attributes of wheat, such as protein concentration and baking properties. Furthermore, the interaction between wheat phenology and environmental factors like temperature and water availability is crucial in determining crop performance and quality. Modulating phenology to optimize grain yield and quality is therefore essential. To gain insights into these adaptation strategies, this research analyzed the outcomes of replicated field trials involving ten wheat varieties, two irrigation regimes, and two sowing dates conducted in the Lleida region of Spain during the 2020–2021 and 2021–2022 growing seasons. This study assessed their impact on wheat phenology, grain yield, yield components, protein concentration, and baking quality parameters. First, this study analyzed the interaction of environmental variables during the two critical wheat developmental stages—vegetative and grain filling—to determine how different varieties with varying phenology are exposed to environmental conditions during these key phases. Second, it quantified temperature and water availability for the different varieties at critical developmental stages. Finally, this study evaluated whether phenology influences grain quality parameters. The outcomes of this research will help refine selection criteria in breeding programs aimed at improving wheat adaptation to climate change.

## 2. Materials and Methods

### 2.1. Experimental Conditions and Plant Material

Replicated trials with ten wheat varieties, two irrigation regimes, and two sowing dates were executed in the Lleida region, Spain, spanning the 2020–2021 (designated as the first year, 2021) and 2021–2022 (designated as the second year, 2022) growing seasons (Table 1). Trials were sown at 400 seeds m^−2^ rates with a plot size of 8 m × 1.2 m. In the initial year, experimentation occurred in Almacelles (coordinates: 41°43′54″ N, 0°25′24″ E, elevation: 221 m), while in the subsequent year, it was conducted in Sucs (coordinates: 41°41′41″ N, 0°25′35″ E, elevation: 284 m). These sites are situated near Lleida, approximately 4 km apart, and share a typical Mediterranean climate, characterized by an average total annual rainfall of 280.7 mm and an annual reference potential evapotranspiration of 1074 mm. In the first year, precipitation and potential evapotranspiration amounted to 364 mm and 1043 mm, respectively, whereas in the second year, they totaled 271 mm and 1105 mm, respectively. All trials were optimally fertilized, and diseases and pests were controlled with locally approved pesticides. In the 2021 and 2022 crop cycles, no nitrogen (N) fertilizer was applied due to the soil containing over 250 kg ha^−1^ of N (due to overfertilization in the past). However, in the 2022 crop cycle, 27 kg ha^−1^ of N were applied before sowing using diammonium phosphate (18–46). Ten distinct commercial varieties were cultivated, deliberately chosen for their contrasting phenotypic traits in phenology. 

All ten commercial varieties of wheat (*Triticum aestivum* L.) underwent evaluation under two distinct irrigation regimes: (i) 100% crop evapotranspiration (ETc), involving irrigation applied at a rate equivalent to one hundred percent of the ETc throughout the growing season, and (ii) rainfed, where no irrigation was administered. Irrigation scheduling for both years was determined using a water balance model using a regional tool for irrigation recommendations: https://ruralcat.gencat.cat/web/guest/eines/eina-recomanacions-de-reg-agricultura (accessed on 14 October 2024). ETc was computed as the product of the reference potential evapotranspiration (ETo), calculated using the Penman–Monteith method [29] and the crop coefficients (Kc) approach, derived from FAO-56, and adjusted based on prevailing weather conditions before being utilized for ETc computations. Sprinklers were installed in the 100% ETc irrigation treatment at an 18 × 18 m grid spacing with a water flow discharge rate of 7.8 L/h/m^2^. Irrigation was scheduled on a weekly basis using a lateral Rainger sprinkler system. 

Additionally, a set of 26 field trials conducted between 2018 and 2023 was used to validate the results obtained from the initial set of ten varieties, irrigation treatments, and sowing dates above. This additional set of field trials included a network of post-registration variety testing trials in Spain, https://extensius.cat/xarxes-de-varietats/ (accessed on 14 October 2024) to provide farmers with annual information on the most adapted varieties of various arable crops. These trials evaluate approximately 20–38 new wheat varieties annually against established benchmark check varieties (“Artur Nick” for spring wheat and “Nogal” for winter wheat) widely cultivated in the region. The replicated trials were conducted using experimental micro-plots (8 m × 1.2 m) located in the most representative production areas across different agroclimatic zones, including Solsona, Artesa de Segre, Olius, and Lleida (Sucs). These areas are characterized by a Mediterranean climate, with hot summers and mild winters. Furthermore, a set of 22 widely grown European wheat varieties was used for validation, grown under the same conditions as the main experimental trials of ten varieties but with only one sowing date. Moreover, two collections of 147 and 158 traditional Mediterranean wheat landraces and commonly grown Mediterranean wheat varieties were used for validation. These trials followed a non-replicated augmented design with two replicated checks (cv. “Anza” and “Soissons”) at a ratio of 1:4 between checks and tested genotypes in 3.6 m^2^ plots with eight rows spaced 0.15 m apart. All these validation trials were optimally fertilized, and diseases and pests were controlled with locally approved pesticides.

### 2.2. Agronomic Traits and Wheat Grain Quality Parameters

For each variety and trial, agronomic and baking quality traits were collected and measured: Days to heading (DH) as the number of days from the sowing date to when 50% of the spikes have emerged on 50% of all stems; days to physiological maturity (DM) as the number of days from the sowing date to when 50% of the peduncles are yellow colored on 50% of all stems; and growing degree days to heading (GDD(DH)) represented the accumulated heat from sowing to heading, calculated by summing the daily average temperatures for each day where the temperature exceeds 0 °C. Similarly, growing degree days to maturity (GDD(DM)) represented the accumulated heat from sowing to maturity, determined by summing the daily average temperatures for each day with a temperature above 0 °C; grain filling duration (GF) as the number of days that occurred between DH and DM (calculated by subtracting DH from DM); grain yield (GY in t ha^−1^) at 13% of humidity was determined by machine harvesting the whole plot; hectoliter weight (HLW in kg/hl) was determined by weighing a 550 mL volume of grains; thousand kernel weight (TKW in g/1000 kernels) was measured using three random samples of 200 whole grains each, with all aborted and broken grains removed; the number of grains m^−2^ (NG) was calculated using the grain yield (GY) and the TKW; protein concentration (Prot %) by using NIR machine (Infratec^TM^ 1241 Grain Analyzer) using minimum 400 g of grains; sedimentation test sodium dodecyl sulfate (SDS in ml) as indicated by Peña et al. [30]. Evaluation of quality and flour rheology was carried out using the Chopin alveograph following the instructions provided by the manufacturer (http://www.kpmanalytics.com/brands/chopin-technologies, accessed on 15 May 2024) to assess dough strength (W in 10^−4^ J), tenacity (P in mm), extensibility (L in mm), and tenacity/extensibility (P/L) ratio. 

### 2.3. Weather Parameters

Weather variables, such as minimum (Tmin), average (TA), and maximum (TM) temperatures (in °C, degree Celsius), precipitation (PP in mm), global solar radiation (SR in MJ m^−2^), and potential evapotranspiration (ETo in mm), were also analyzed. Meteorological data were collected from an automated weather station located 3 km and 5 km from the Almacelles and Sucs study sites, respectively. The following sensors were connected to an AgDevice data logger (AgZoom): Rainfall was measured using an ECRN-50 rain gauge with 0.25 mm resolution; solar radiation was measured with a pyranometer (model CM11, Kipp and Zonen Delft, Holland); air temperature, humidity, and air pressure were measured using the Decagon VP-4 system (Decagon Devices); and wind speed was measured with a Davis Cup4 Anemometer (Decagon Devices). Meteorological data for the additional 26 trials were obtained from the official network (SMC, http://www.ruralcat.net/web/guest/agrometeo, accessed on 1 August 2023), equipped with the specified devices available at: https://www.meteo.cat/wpweb/divulgacio/equipaments-meteorologics/estacions-meteorologiques-automatiques/xarxa-destacions-meteorologiques-automatiques-xema/els-sensors/, accessed on 7 August 2024). Temperatures, along with other weather parameters, were recorded on an hourly basis. Mean maximum temperature (TM) was calculated as the average of the highest temperatures recorded each day, calculated over each month. Mean minimum temperature (Tmin) was calculated as the average of the lowest temperatures recorded each day, calculated over each month. Average temperature (TA) was the average of all the temperatures recorded over each month. Subsequently, averages and cumulative values were separately computed for the vegetative and grain filling stages to assess the relative exposure to these weather variables during the two different phases of the plant cycle in each wheat variety. This process involved the following methods: Averaged values (for temperature variables) were derived, for the vegetative stage, by dividing the sum of daily measurements from sowing to heading date by the number of days in this period (days to heading); for the grain filling stage, the sum of daily measurements from heading to physiological maturity date was divided by the number of days in this period (grain filling duration). Cumulative values (for PP, SR, and ETo) were calculated by summing the daily measurements for each specified period. Consequently, all these newly calculated variables were designated by adding “V” for those calculated during the vegetative stage and “GF” for those calculated during the grain filling stage. In calculating the cumulative potential evapotranspiration for each variety, it was assumed that the main differences between varieties in terms of exposure to temperature, precipitation, and evapotranspiration were primarily due to the duration and extent of the crop cycle. The intrinsic differences in evaporation and transpiration due to biomass variations in different wheat varieties were considered negligible when compared to the overall water balance over time throughout the different crop cycles of each variety. This has been demonstrated and supported by Gómez-Candón et al. [31], who found that the actual evapotranspiration of each variety is closely correlated with the reference evapotranspiration. Finally, total ETo from sowing to maturity for each variety was calculated. 

### 2.4. Statistical Analysis

Randomized Complete Block Design (RCBD) was used to conduct all the replicated trials employed in this study. For each trial and season, the effects of year, sowing date, water, and variety and all their possible interactions were tested using analysis of variance (ANOVA) applying a linear statistical model considering all these factors as fixed. Student’s means separation test was performed to detect any significant separations in 2 groups between irrigated and rainfed, first (1 SD) and second (2 SD) sowing dates, 2021 and 2022, while for variety was performed Tukey’s test. Simple linear regression analysis was performed using all the data (sowing dates, irrigation, years, and varieties) and using the averages of each variety across all environments and per irrigation and sowing date. Correlation analysis was conducted for agronomic traits and weather parameters during the vegetative and grain filling stages of each variety. The statistical software package SAS-JMP Pro 16 (SAS Institute Inc., Cary, NC, USA, 1989–2019) was used to perform all the reported statistical analyses. Graphs reporting the correlations and regression equations were plotted using Microsoft Excel 365 (Version 2304, Redmond, WA, USA).

## 3. Results

### 3.1. Weather Profiling Across Two-Year Experimental Trials

The winter of 2021, in comparison to 2022, was marked by elevated temperatures (1.9 °C and 3.4 °C more during 2021 in January and February Tmin with long-term averages of 0.8 °C and 1.3 °C, respectively) and a cooler spring–summer period (5.5 °C and 5.0 °C less during 2021 in May and June TM with long-term averages of 24.6 °C and 29.7 °C, respectively). From January to June 2021, the total cumulative precipitation and potential evapotranspiration were 38.3 mm higher and 21.2 mm lower, respectively, than in 2022. Overall, the weather in 2022 was warmer and drier than in 2021 (Table 2). Moreover, the weather parameters during the two crop cycles (2020–2021 and 2021–2022) and the phenology fit of each variety within the two years are shown in Figure 1. 

This plot visually illustrates the genotype-dependent variability in phenology, resulting in differing temperature and water availability exposure during the vegetative and grain filling stages of the different varieties (Figure 1).

### 3.2. Effects of Year, Sowing Date, Water Regime, and Variety on Agronomic and Wheat Grain Quality Properties

A four-way factorial ANOVA was conducted using “year”, “sowing” (date), “water” (regime), and “variety” as factors along with their respective interactions. Appendix A presents the sum of squares (SS) for each factor, expressed as a percentage of the total explained variation for each variable. For the grain quality traits, a three-way factorial ANOVA was performed, removing variety factors and all their interactions (Appendix A), showing year as the most significant factor. Mean separation tests were performed to detect the differences between levels of the main factors (irrigated against rainfed and sowing dates, one SD against two SDs) and between the two years of trials (Table 3). Mean separation tests, based on differences in water availability, sowing dates, and years (2021 and 2022), were consistently significant. Specifically, under irrigated conditions, compared to rainfed, there were higher values for GDD(DH), GDD(DM), GF, GY, TKW, and NG (not significant for HLW). Conversely, under rainfed conditions, they reported higher values of Prot (%) and SDS than well-irrigated ones.

Regarding grain quality traits, a significant increase in W and P was observed in the rainfed treatment compared to the well-irrigated one. Regarding the effects of sowing date, late sowing (two SDs) resulted in decreased GDD(DH), GDD(DM), GF, GY, HLW, TKW, and NG, while grain protein concentration and SDS were increased in the late sowing. Late sowing did not significantly affect the quality parameters of W and P/L. However, in 2022 (a warmer and drier year), there was a decrease in W, P, and L compared to the observations made in 2021.

### 3.3. Impact of Year, Sowing Date, Water Regime, and Variety on Average Crop Weather Exposure During the Vegetative and Grain Filling Stages

Water regime and variable sowing dates significantly influenced crop phenology (Table 3), leading to variations in growth conditions during the two major crop growth stages: vegetative and grain filling. 

A four-way factorial ANOVA was additionally conducted utilizing crop exposure weather variables (refer to Appendix A). Wheat varieties grown under full irrigation consistently experienced increased temperatures, precipitation, global solar radiation, and potential evapotranspiration (ETo) during both the vegetative and grain filling stages and ETo TOT from sowing to maturity (Table 3). This was primarily attributed to the 6-day difference in days to maturity between the irrigated and rainfed regimes since irrigation extends the crop cycle. Furthermore, a consistent temperature difference was observed between the two sowing dates, with the first sowing date experiencing lower temperatures compared to the second. Precipitation levels were higher for the first sowing date during both vegetative and grain filling stages. However, global solar radiation and potential evapotranspiration were lower in the first sowing date (compared to the late sowing 2SD) at the vegetative stage but higher at the grain filling stage. (Table 3). In 2021, lower temperatures were recorded during the vegetative stage (except Tmin, which showed higher values in 2021), and higher values were reported for all other variables during both the vegetative and grain filling stages, including precipitation, global solar radiation, and potential evapotranspiration.

### 3.4. Investigating the Linear Impacts of Temperature, Water Availability, and Potential Evapotranspiration During the Vegetative and Grain Filling Stages on the Agronomic and Grain Quality Characteristics of Wheat Grown Under Two Sowing Dates and Two Water Regimes (Major Environmental Effects)

Simple linear regression was performed for both GY and Prot (%) with ETo and TM during grain filling and total ETo from sowing to maturity (Figure 2A–F). ETo GF showed significant positive correlations with GY (slope = 0.13 and R^2^ = 0.85) and with Prot (%) (slope = −0.06 and R^2^ = 0.45). TM GF showed a significant negative correlation with GY (slope = −0.84, R^2^ = 0.51) and a weakly positive correlation with Prot (%) (slope = 0.17, R^2^ = 0.05). 

When considering total ETo from sowing to maturity, grain yield was positively correlated with R^2^ = 0.22 (Figure 2E). While the correlations between yield and weather conditions during the vegetative or grain filling stages were consistent both across environments (dashed regression line) and across varieties (solid lines) in Figure 2, the genetic response to total ETo showed a contrasting pattern, which will be discussed further in Section 3.5.

Correlation analysis was performed between agronomic and quality traits with environmental conditions (including all 80 data points across environments and varieties) during vegetative and grain filling growth stages (as summarized in Table 4). The traits most strongly correlated with GY and its components were minimum temperature (Tmin), average temperature (TA), maximum temperature (TM) during grain filling, precipitation (PP during GF), solar radiation, and evapotranspiration (Table 4). Grain quality properties (P/L and W) were also affected by temperature, precipitation, solar radiation, and ETo, particularly during grain filling (Table 4).

### 3.5. Investigating the Impacts of Temperature, Water Availability, and Potential Evapotranspiration During the Vegetative and Grain Filling Stages on the Agronomic and Grain Quality Characteristics of Ten Wheat Varieties Grown Under Two Sowing Dates and Two Irrigation Regimes (Genetic Effects)

Tukey’s mean separation test was performed among varieties for agronomic traits, environmental conditions during the vegetative and grain filling stages, and quality traits (Table 5). Despite cultivating the ten varieties under uniform conditions regarding water regimes and sowing dates, inherent variations in variety phenology led to differences in the timing of vegetative and grain filling stages among the ten varieties (see Table 5 and Appendix A). The differences in the timing of vegetative and grain filling stages among the ten varieties had significant implications for the temperature and water availability to which the ten varieties were exposed. Furthermore, the earliest varieties experienced an average maximum temperature of 14.4 °C during the vegetative phase and 26.8 °C during the grain filling phase. Yields consistently declined at temperatures above these temperatures, as demonstrated in Table 5, which includes data from the earliest variety (var. 4) to the latest variety (var. 5) and all varieties in between.

Significant variation was observed for all traits and environmental conditions across varieties. Given that each variety was subjected to varying average temperatures, potential evapotranspiration, and precipitation levels during both the vegetative and grain filling stages, assessment of the genetic-level impact of these differences was undertaken. The correlations between GY and key environmental factors, including ETo, protein concentration (Prot %), and TM, during both the vegetative and grain filling stages across multiple trials were significant across the ten varieties using the means of each variety across experimental conditions and years (Table 6). Moreover, a consistent decrease in GY with increasing ETo V was observed, with correlations ranging from −0.80 to −0.95. Among yield components, HLW resulted more negatively correlated with ETo (with correlations ranging from −0.71 to −0.92), while TKW and NG were less negatively correlated with ETo V (Table 6). Conversely, positive correlations between ETo V and protein concentration were observed, ranging from 0.74 to 0.96, though in three trials (once averaging the two years) these correlations were not significant (Table 6). When considering the total ETo (ETo TOT) from sowing to maturity of each variety, correlations with yield became negative (Table 6). Temperature during the vegetative stage exhibited a similar pattern, with negative correlations with GY ranging from −0.81 to −0.95 and positive correlations with grain protein concentration ranging from 0.74 to 0.96.

Per-trial analysis for the grain filling stage showed strong positive correlations between GY and ETo GF, ranging from 0.67 to 0.96, except for one trial where it was not significant. Grain protein concentration exhibited consistent negative correlations with ETo GF, ranging from −0.81 to −0.96 across all trials except in Sucs 2022 1SD Irrigated (Table 6). Increased temperature negatively impacted GY, as shown by correlations of GY with TM GF (negative and significant, ranging from −0.75 to −0.91). Moreover, HLW consistently showed negative correlations with TM GF ranging from −0.74 to −0.91 (only once not significant). To validate the dependence of temperature during the grain filling stage on HLW, attributable to phenological differences among wheat varieties, a series of 26 trials conducted over different years were analyzed. A negative correlation between HLW and TM V was observed in 16 out of 26 trials, while a negative correlation between HLW and TM GF was reported in 8 out of 15 trials (Appendix A). 

Overall, when considering the total ETo (ETo TOT) from sowing to maturity for each variety, the correlation with yield became negative. Additionally, the average temperature from sowing to maturity also maintained a negative correlation with yield (Table 6).

Correlation analysis was also conducted to examine the relationship between quality traits and weather variables, specifically potential evapotranspiration and maximum temperature. In this case, the analysis was carried out on a per-trial basis to verify the genetic effects attributable to phenological differences among wheat varieties (Table 7). Considering the genetic effects, ETo V and TM V showed negative correlations with grain quality parameters P/L and P, while exhibiting positive correlations with L. Remarkably, no significant correlations were found between dough strength (W) and weather variables during both vegetative and grain filling stages (refer to Table 7) when using the means of each variety. 

The effect of temperature during the grain filling stage on quality properties was also significant in four out of eight trials with negative correlations observed for P/L (ranging between −0.72 and −0.80, Table 7). Half of the trials exhibited negative correlations between TM GF and P, whereas positive correlations were observed for L across five out of eight trials, ranging from 0.67 to 0.74. When considering the average temperature from sowing to maturity (TM TOT) of each wheat variety, consistent negative effects on P/L were observed, except for the 2022 SUCs 2SD Rainfed trial (Table 7). 

### 3.6. Correlation Between Hectoliter Weight and Quality Traits

Correlation analysis per trial was conducted between HLW and quality traits. P/L and P showed high and significant positive correlation values with HLW, ranging from 0.67 to 0.94 and 0.65 to 0.93, respectively (except in 2022 Sucs 2SD Rainfed and 2022 Sucs 1SD Irrigated, where it was non-significant, respectively, Table 8). Consistently significant negative correlations were also observed between HLW and L, with values ranging from −0.74 to −0.83 (again, except for one instance in 2022 Sucs 2SD Rainfed, where it was non-significant, Table 8). HLW was not correlated with W, except when means of two years were calculated for Rainfed 2SD (Table 8). 

## 4. Discussion

### 4.1. Effects of Late Sowing, Water Restrictions, and Interannual Weather Variation on Wheat Grain Yield, Yield Components, and Grain Quality 

Many studies highlight the significant negative impact of combined drought and heat stress on crop yield, yield components, and the positive impact on grain protein [6,28,32]. Additionally, research has shown that late sowing of crops, often due to climate-induced factors such as delayed harvesting or rainfall, can lead to substantial yield reductions, particularly affecting wheat and rice production [33,34]. Relatively limited research exists concerning the effects of temperature and drought stress on the quality attributes of wheat grain. Specifically, Li et al. [35] investigated the effects of drought and heat stress on wheat quality parameters and found that under stress conditions, flour protein and SDS sedimentation significantly increased compared to non-stress conditions. This increase in protein concentration may be due to higher rates of grain nitrogen accumulation and/or lower rates of carbohydrate accumulation. Li et al. [35] demonstrated that an increase in P and a decrease in L led to higher tenacity (a larger P/L ratio), which could explain the reduced loaf volume observed under drought stress. Conversely, a decrease in P and an increase in L might account for the enhanced bread volume seen under heat stress (approximately a 1 °C increase from late sowing). Despite these changes in quality parameters, heat and drought stress led to more severe reductions in grain yield compared to quality traits, particularly considering that only a 1 °C increase in temperature was observed [35]. The present study confirmed that temperature and water scarcity exerted significant negative effects on yield. However, wheat grain quality was less affected, at least in the region and years where this was tested (with only W and P showing significant increases of 27% and 19%, respectively, attributed to reduced water availability). This is especially true under the relatively small variations in temperature and water availability observed in the experimental conditions of late sowing and rainfed cultivation within each year, compared to local common sowing practices and supplementary irrigation. However, it is important to note that the yield reported for the plot size used in this study (8 m × 1.2 m) may not fully represent actual yields in full-sized fields [36]. While the comparative effects across treatments might be reliable, the actual yields observed in these plots are not necessarily indicative of those from larger, full-scale fields. 

Elevated temperatures adversely affect grain yield through several interconnected mechanisms. High temperatures can directly inhibit grain filling and related processes, leading to reduced yields. Additionally, increased temperatures commonly shorten the grain filling period and accelerate crop senescence, reducing the time crops have to capture solar radiation, absorb CO_2_, and produce dry matter. The impact of elevated temperatures also varies among genotypes, as different varieties have distinct responses to heat stress. Furthermore, factors like crop cover, which affects how much solar radiation is intercepted, and the ability to translocate vegetative dry matter to grain differ among varieties, further influencing yield outcomes under high-temperature conditions [5,12,13,14,15,16,17]. In this study, correlation analysis of the 10 wheat varieties (Table 5) revealed strong correlations between yield and growing degree days, yield and grain filling duration, and yield with TM, ETo, and SR during the vegetative and grain filling stages. These findings align with previous results, showing that higher temperatures accelerate crop development, shorten the period for absorbing solar radiation and undergoing evapotranspiration, and ultimately result in lower yields. Considering the increased variability and impact of interannual weather fluctuations, with a temperature increase of 5 °C in 2022 compared to 2021 and 38 mm less precipitation during the grain filling period in 2022 than in 2021, the observed impacts on grain yield and quality parameters were more pronounced (with a decrease of 51%, 55%, 43%, and 24% for grain yield, W, P, and L, respectively). The significant interannual weather variations led to negative correlations between dough strength (W) and tenacity (P) with maximum temperature over the two-year period. These results indicated that temperatures during grain filling exceeding 25 °C up to 30 °C (with a 5 °C range of variation) can adversely influence both W and P (with Pearson values of −0.75 and −0.82, respectively, for the correlations between W and TM and between P and TM across years, Table 4). This is consistent with previous findings that indicate protein concentrations increase above 30 °C, but this is accompanied by a decline in dough strength [23,24]. Polymeric/monomeric ratio and molecular size in gluten composition can play an important role in conferring dough strength. Alterations on the gliadin/glutenin ratio, which is generally 1:1 even if dependent on genotype and growth conditions, can be caused by high temperature exposition, which disturbs aggregation properties [37,38,39,40]. At the same time, as the ratio gliadin/glutenin increases in conditions of high temperature (at 30–35 °C), the ratio of large to small polymers decreases [18,20]. Moreover, the decline in glutenin synthesis in heat stress conditions leads to a reduction of disulfide cross-linking among glutenin subunits, resulting in poor dough quality [18]. Nevertheless, Blumenthal et al. [18] reported that not all the genotypes were responding to heat stress in the same way, identifying tolerant and more susceptible genotypes. On the other hand, cultivars with high gluten strength were significantly correlated to a high total amount of glutenins, a high amount of HMW-GS, and a high Glu/Gli ratio ([19], as reviewed by Ullah et al. [13]).

Peña et al. [30] demonstrated that variations in protein concentration, influenced by both genetic and external factors, can significantly impact dough strength. To test the impact of genetic differences in grain quality properties in the present study, further analysis of W and P/L was conducted among varieties. Specifically, during the grain filling stage, temperatures varied by an average of 3.3 °C (with the latest varieties being exposed to more than 30 °C) between the earliest and latest varieties (Table 5), indicating potential effects on quality due to phenology. This analysis is detailed in the next discussion Section 4.3.

### 4.2. Exploring Genetic Diversity and Environmental Conditions Across Growth Stages in Wheat Varieties with Distinct Phenological Characteristics in Field Trials—Grain Yield

Early cereal genotypes yield more under terminal drought conditions, leveraging a drought escape mechanism [41,42,43,44]. Meiosis, a critical and stress-sensitive stage in wheat reproduction, typically aligns with the booting stage and is crucial for avoiding drought and heat to ensure survival and grain set [45,46,47]. Blum [47], referencing Mitchell et al. [42], noted that short-duration genotypes conserve water and avoid terminal drought stress, whereas long-duration genotypes, with their extensive root systems, can utilize deep soil moisture if available. This phenological escape strategy, however, results in early wheat varieties being exposed to different temperature and water availability conditions at specific developmental stages, compared to late varieties. Consequently, comparing genotypes with different crop development durations under the same drought or heat conditions would likely reveal varying effects on grain yield due to differences in weather exposure. To delve deeper into these hypotheses, this study revealed significant differences in potential evapotranspiration, solar radiation, and temperature exposure among the early and late wheat varieties. Specifically, early varieties experienced lower potential evapotranspiration and temperatures during the vegetative stage; however, they were exposed to higher evapotranspiration and lower temperatures during the grain-filling stage. Additionally, early varieties exhibited an extended grain-filling period, averaging about 9 days longer than later varieties. The lower temperatures during grain filling (3.3 °C lower) and the longer grain-filling duration, combined with increased ETo during grain filling (ETo GF), likely contributed to greater total photosynthetic activity and increased availability of photo-assimilates for grain filling and yield formation. With this strategy, early varieties may keep photosynthesis active during the most crucial period of grain filling, maximizing yield formation. Despite the extended grain filling duration, the total ETo from sowing to maturity in early varieties was lower than in late varieties, resulting in significant water savings throughout the crop cycle, particularly advantageous in water-limited conditions (Table 5). Together, these results may explain the higher yields observed in early varieties in this study. These findings align with the widely accepted concept that temperature and solar radiation are key weather factors influencing yield variability across wheat varieties. Dr. Fischer [48] emphasized this by demonstrating that yield potential fluctuations are strongly linked to the photothermal quotient (the ratio of solar radiation to mean temperature) in the month leading up to flowering, which affects kernel number, and to the temperature during grain filling, which plays a crucial role in determining kernel weight [48]. Further research may be needed to estimate the optimal temperature for phenology, specifically the point at which developmental phases are shortest and beyond which development and related factors become limited. Additional data, especially from even earlier-maturing varieties than those used herein, would be essential to accurately determining this threshold.

### 4.3. Exploring Genetic Diversity and Environmental Conditions Across Growth Stages in Wheat Varieties with Distinct Phenological Characteristics in Field Trials—Grain Quality

The next level of analysis aimed to determine whether the differences in environmental exposure between early and late varieties had any impact on the final grain quality properties. It was observed that the phenology of the varieties was directly linked to grain quality properties, as demonstrated by the consistent negative correlations between growing degree days to heading and both hectoliter weight (HLW) and P/L ratio (Appendix A). To our knowledge, this study is the first to show a direct correlation between phenology, temperature exposure, and grain quality. Hectoliter weight, a long-established wheat grading specification, serves as a reliable indicator of wheat’s physical quality and flour yield due to its ease of measurement, affordability, and consistent results [49]. Our findings demonstrated a significant effect of temperature during both the vegetative and grain filling stages on hectoliter weight. This was further validated through a series of 26 trials, with significant correlations observed in over 50% of these trials (Appendix A). This effect was observed through variations in environmental conditions related to sowing dates (Table 3) and differences among varieties in phenology, resulting in varying temperature exposure. Higher temperatures in late varieties decreased hectoliter weight and the P/L ratio, with very little effect on W. The P/L ratio represents the balance between tenacity and extensibility, while W indicates the overall energy required to deform the dough, reflecting its baking strength. Consequently, the correlations between HLW and P/L, P, and L emphasize the relationship between phenology and grain quality under climate change scenarios. Li et al. [35] demonstrated a decrease in loaf volume under heat stress related to a decrease in P and an increase in L; however, their analysis did not include phenology and the linkages between different varietal phenology, temperature exposure during grain filling, and wheat quality properties were not considered. The results presented herein highlight, for the first time, the importance of phenology on wheat quality of different varieties, suggesting that targeted selection for specific phenology could mitigate the negative impacts of heat stress not only on grain yield but also on grain quality. 

However, it is important to note that the adaptation strategy of shorter vegetative stages in Mediterranean-type environments is not the only genetic approach available. For example, Liu et al. [16] found through simulations that wheat varieties with extended juvenile phases performed better under late sowing conditions. While these results may appear to contradict those presented here, they highlight that adaptation strategies can be specifically tailored to suit different environments and cropping systems. 

Overall, the results presented here suggested that in Mediterranean environments, optimizing wheat grain yields requires aligning phenology to minimize exposure to high temperatures during grain filling while maximizing evapotranspiration potential through an extended grain filling period to enhance photosynthesis. Simultaneously, this strategy also reduces total evapotranspiration from sowing to maturity, leading to greater water savings, which is especially beneficial under water-limited conditions. While this study showed the benefits of early-maturing wheat varieties in regions prone to terminal drought and heat, such as Mediterranean climates, it also unveiled a significant concern. It was observed that early varieties were exposed to different environmental conditions when compared to late varieties, showing different timings in terms of the vegetative and grain filling stages. This discrepancy raises a crucial question about the criteria used to designate a wheat variety as drought- or heat-tolerant. Specifically, when evaluating tolerance, the timing of heading and maturity must be considered to ensure both varieties are exposed to identical environmental conditions for a fair comparison. This substantially calls into question the methods used to identify drought and heat tolerance genotypes (in terms of grain yield and quality) and highlights the importance of evaluating phenological groups when deciding about the levels of tolerance of certain wheat varieties. 

## 5. Conclusions

This study explored the effects of late sowing, water restrictions, and interannual weather variation on wheat grain yield, yield components, and grain quality across different wheat varieties. The findings reveal that high temperatures and drought stress negatively impact wheat yield while also influencing grain quality attributes. Late sowing, which often results in exposure to higher temperatures, significantly reduced grain yield, but the quality traits showed a more variable response, with positive effects on protein concentration. Early-maturing (compared to late-maturing) varieties demonstrated a distinct advantage under these challenging environmental conditions. Specifically, they experienced lower potential evapotranspiration and temperatures during the vegetative stage, followed by increased evapotranspiration and cooler temperatures during grain filling. This extended grain-filling period (about 9 days longer on average than later varieties) and cooler conditions during grain filling were conducive to higher solar radiation interception and potentially higher photosynthetic activity, resulting in greater grain yield. Additionally, early varieties saved water over the crop cycle from sowing to maturity, which was particularly beneficial in water-limited environments. This study also highlighted the critical role of phenology in determining grain quality. Consistent negative correlations between temperature and hectoliter weight and the P/L ratio across wheat varieties confirmed the impact of temperature on grain quality properties. These findings suggest that future rising temperatures could deteriorate the quality of wheat for bread-making, impacting the baking industry, consumer satisfaction, and global food security. This study emphasizes the urgent need for developing adaptive agricultural practices and breeding heat-tolerant wheat varieties to mitigate the effects of climate change.

## Figures and Tables

**Figure 1 plants-13-02929-f001:**
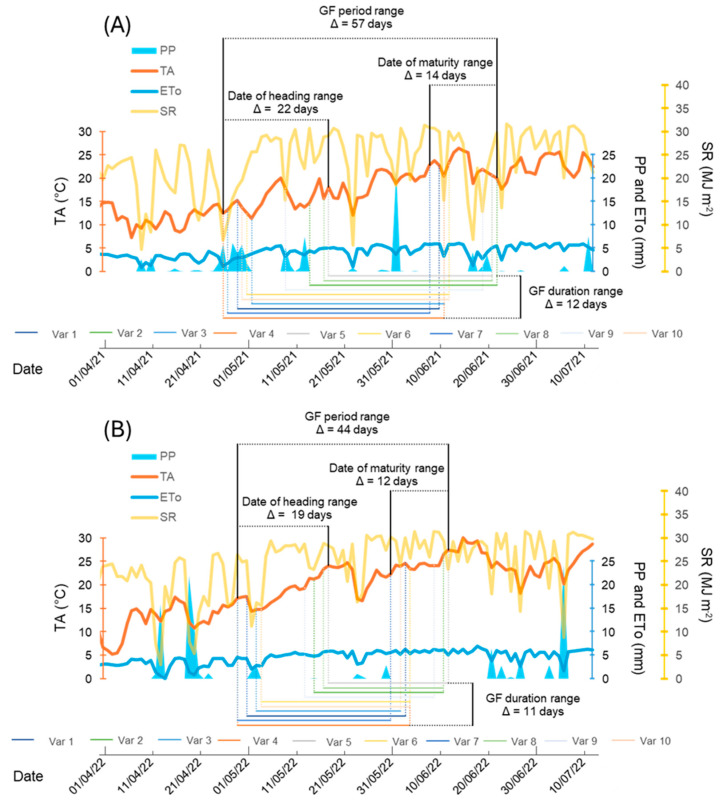
Average temperature (TA, °C), precipitation (PP, mm), potential evapotranspiration (ETo, mm), and global solar radiation (SR, MJ m^−2^) during grain filling in each wheat variety (var. 1 to 10) in (**A**) 2020–2021 Almacelles Lleida NE Spain, and (**B**) 2021–2022 Sucs Lleida NE Spain crop cycles. Phenology dates, such as date of heading, date of maturity, and grain filling duration (GF) ranges, are visually represented. The time span on the x axes of both panels is indicated by a line with major tick marks every month and by minor tick marks every 10 days.

**Figure 2 plants-13-02929-f002:**
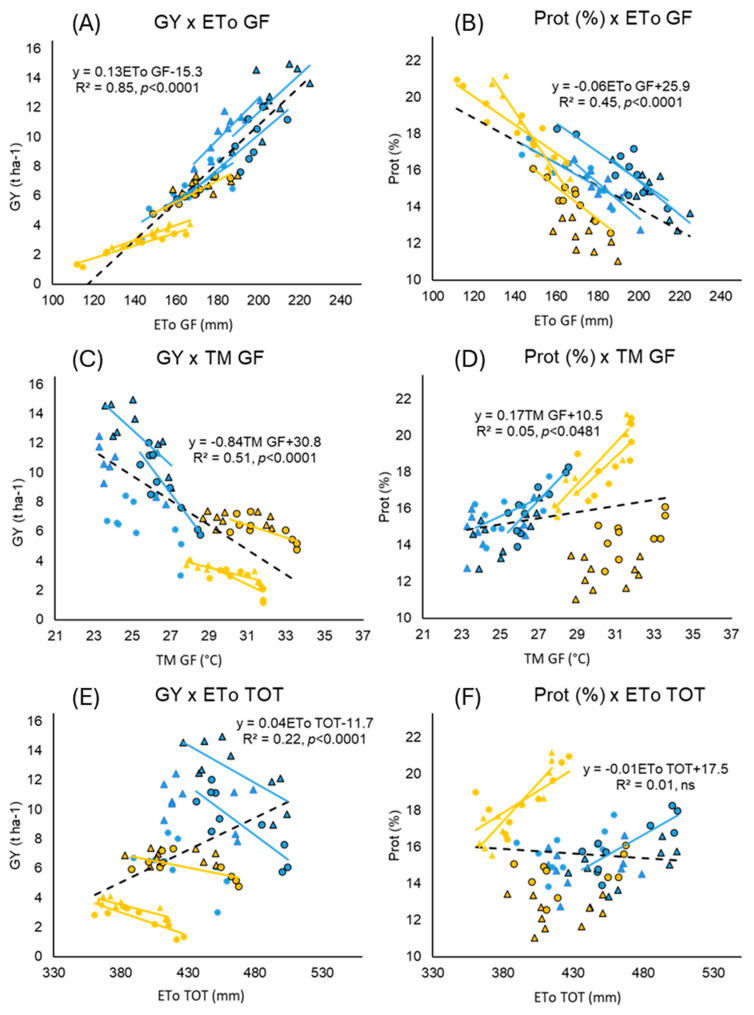
Simple linear regression between weather parameters experienced by each wheat variety during grain filling, considered as the time between heading and maturity (cumulative potential evapotranspiration, ETo, maximum temperature exposure, TM, and total potential evapotranspiration, ETo TOT), grain yield (GY, panels (**A**,**C**,**E**)), and protein concentration (Prot %, panels (**B**,**D**,**F**)). Linear regressions for each year and experimental condition are displayed with a solid line if they are statistically significant.

**Table 1 plants-13-02929-t001:** The dates of sowing (SD) for each experimental trial (indicated with location, year of harvest, first or second sowing date, and 100% irrigated or rainfed), as well as the number of days between two sowing dates (Δ sowing date) in each year of trials, are shown.

Trial	Sowing Date	Δ Sowing Date
Almacelles 2021 1SD Irrigated	3 December 2020	
Almacelles 2021 2SD Irrigated	27 December 2020	24
Almacelles 2021 1SD Rainfed	3 December 2020	
Almacelles 2021 2SD Rainfed	27 December 2020	24
Sucs 2022 1SD Irrigated	7 December 2021	
Sucs 2022 2SD Irrigated	31 December 2021	24
Sucs 2022 1SD Rainfed	6 December 2021	
Sucs 2022 2SD Rainfed	31 December 2021	25

**Table 2 plants-13-02929-t002:** Monthly weather parameters from winter to summer (from January to June) in Lleida include minimum, average, and maximum temperatures (Tmin, TA, and TM), cumulative precipitation (PP), global solar radiation (SR), and potential evapotranspiration (ETo). The averages for temperature and the total cumulative precipitation, global solar radiation, and potential evapotranspiration are presented in the final row in each crop cycle (2021 and 2022).

	Tmin (°C)		TA (°C)		TM (°C)		PP (mm)		SR (MJ m^−2^)		ETo (mm)	
	2021	2022	2021	2022	2021	2022	2021	2022	2021	2022	2021	2022
January	−0.9	−2.8	3.7	2.8	9.1	10.1	50.5	4.0	237.3	289.8	24.6	27.9
February	5.1	1.7	9.7	8.1	14.9	15.6	21.5	3.3	268.5	327.6	35.9	42.3
March	3.0	5.2	9.8	10.0	17.3	15.5	9.3	39.2	538.5	333.8	77.5	50.9
April	6.3	6.2	12.0	12.8	18.4	20.0	30.3	52.8	559.8	590.1	86.3	91.4
May	9.8	11.8	16.7	19.9	23.6	29.1	17.8	9.0	805.6	790.2	138.8	149.6
June	14.8	16.8	21.7	24.9	28.9	33.9	31.1	13.9	750.5	817.8	144.5	166.7
Average/Total	6.3	6.5	12.3	13.1	18.7	20.7	160.5	122.2	3160.2	3149.3	507.5	528.7

Tmin: minimum temperature; TA: average temperature; TM: maximum temperature; PP: precipitation; SR: global solar radiation; ETo: potential evapotranspiration.

**Table 3 plants-13-02929-t003:** Student’s mean separation tests of water regime (irrigated and rainfed), sowing date (two sowing dates), and year (2021 and 2022) effects for agronomic traits, weather variables, and grain quality parameters measured in ten wheat varieties. Significant values are displayed in blue for higher values and in red for lower values when *p* < 0.05. Non-significant differences (*p* > 0.05) are indicated in gray cells.

	Plant Trait Variables					
	Irrigated	Rainfed	1 SD	2 SD	2021	2022
GDD(DH)	**1245**	**1214**	**1243**	**1216**	**1239**	**1221**
GDD(DM)	**1996**	**1847**	**1958**	**1884**	**1988**	**1854**
GF	**37.2**	**32.7**	**37.1**	**32.9**	**40.3**	**29.6**
GY	**8.7**	**5.7**	**8.3**	**6.1**	**9.6**	**4.7**
HLW	75.2	75.3	**76.2**	**74.3**	**77.5**	**72.9**
TKW	**34.5**	**29.6**	**33.3**	**30.8**	**33.5**	**30.6**
NG	**25,309**	**18,439**	**24,355**	**19,392**	**28,668**	**15,080**
Prot (%)	**14.0**	**16.4**	**14.6**	**15.8**	**15.0**	**15.5**
SDS	**12.6**	**14.1**	**12.8**	**13.9**	**11.9**	**14.8**
	Weather variables					
Tmin V	**3.2**	**3.1**	**2.9**	**3.3**	**3.2**	**3.0**
TA V	**8.7**	**8.6**	**8.2**	**9.1**	**8.6**	**8.7**
TM V	**15.0**	**14.9**	**14.3**	**15.6**	**14.6**	**15.3**
PP V	**117.9**	**116.6**	**122.9**	**111.6**	**128.3**	**106.3**
SR V	**1845.8**	**1798.7**	**1806.3**	**1838.2**	**1871.2**	**1773.3**
ETo V	**260.5**	**252.3**	**249.6**	**263.2**	**261.9**	**251.0**
Tmin GF	**12.7**	**12.0**	**12.0**	**12.8**	**11.7**	**13.0**
TA GF	**20.4**	**19.6**	**19.5**	**20.5**	**18.6**	**21.4**
TM GF	**28.6**	**27.8**	**27.6**	**28.8**	**25.7**	**30.7**
PP GF	**25.0**	**24.4**	**28.5**	**20.9**	**43.7**	**5.7**
SR GF	**985.4**	**866.8**	**966.0**	**886.2**	**1045.2**	**807.0**
ETo GF	**184.8**	**159.7**	**178.1**	**166.4**	**187.8**	**156.6**
ETo TOT	**445.3**	**411.9**	427.7	429.6	**449.7**	**407.6**
	Quality traits*					
W	**199**	**252**	216	234	**310**	**140**
P/L	0.65	0.74	0.70	0.69	0.76	0.63
P	**58**	**69**	63	64	**81**	**46**
L	97	107	99	105	**116**	**88**

GDD(DH): growing degree days to heading (°Cd); GDD(DM): growing degree days to maturity (°Cd); GF: days of grain filling (days); GY: grain yield (t ha^−1^); HLW: hectoliter weight (kg/hl); TKW: thousand kernel weight (g/1000 kernels); NG: number of grains m^−2^ (units m^−2^); Prot (%): grain protein concentration (%); SDS: sodium dodecyl sulfate sedimentation (ml). Weather variables are followed by V (during the vegetative stage) and GF (during the grain filling stage). Tmin: minimum temperature (°C); TA: average temperature (°C); TM: maximum temperature (°C); PP: precipitation (mm); SR: global solar radiation (MJ m^−2^); ETo: potential evapotranspiration (mm); ETo TOT: total potential evapotranspiration from sowing to maturity (mm); W: dough strength (10^−4^ J); P/L: tenacity/extensibility ratio; P: tenacity (mm); L: extensibility (mm). * indicates that the separation tests for these variables were performed removing the variety factor because of a lack of replicates.

**Table 4 plants-13-02929-t004:** Pearson correlation coefficients of plant traits and grain quality parameters with weather variables, including in the analysis data from trials in 2021 and 2022, are shown (using N = 80 data points of 10 varieties in eight trials). Significant correlations are indicated in bold font (*p* < 0.05), and normal format indicates non-significant correlations.

	Vegetative Stage						Grain Filling Stage					
Variable	Tmin V	TA V	TM V	PP V	SR V	ETo V	Tmin GF	TA GF	TM GF	PP GF	SR GF	ETo GF
GDD(DH)	**0.78**	**0.64**	**0.47**	**0.46**	**0.96**	**0.95**	**0.72**	**0.53**	**0.36**	−0.12	**−0.32**	**−0.26**
GDD(DM)	**0.58**	**0.34**	0.08	**0.73**	**0.80**	**0.73**	**0.38**	0.14	−0.05	**0.29**	**0.31**	**0.39**
DH	0.18	−0.04	−0.19	**0.59**	**0.60**	**0.50**	**0.27**	0.16	0.08	0.05	−0.03	0.01
DM	0.00	**−0.31**	**−0.53**	**0.78**	**0.39**	**0.26**	−0.13	**−0.28**	**−0.37**	**0.49**	**0.49**	**0.50**
GF	**−0.29**	**−0.53**	**−0.71**	**0.54**	−0.22	**−0.32**	**−0.70**	**−0.81**	**−0.84**	**0.85**	**0.99**	**0.95**
GY	−0.16	**−0.40**	**−0.58**	**0.57**	−0.06	−0.16	**−0.52**	**−0.65**	**−0.72**	**0.82**	**0.94**	**0.92**
HLW	**−0.52**	**−0.60**	**−0.63**	0.12	**−0.55**	**−0.61**	**−0.78**	**−0.75**	**−0.69**	**0.60**	**0.62**	**0.53**
TKW	**−0.24**	**−0.31**	**−0.34**	0.11	−0.22	**−0.25**	**−0.30**	**−0.33**	**−0.33**	**0.40**	**0.55**	**0.56**
NG	−0.05	**−0.31**	**−0.53**	**0.66**	0.06	−0.04	**−0.46**	**−0.61**	**−0.70**	**0.80**	**0.90**	**0.89**
Prot (%)	**0.39**	**0.44**	**0.43**	−0.06	**0.34**	**0.38**	**0.29**	**0.25**	**0.22**	−0.22	**−0.60**	**−0.67**
SDS	−0.06	0.21	**0.44**	**−0.57**	−0.15	−0.06	**0.33**	**0.50**	**0.60**	**−0.70**	**−0.76**	**−0.75**
W	0.05	−0.14	**−0.35**	**0.40**	0.00	−0.06	**−0.56**	**−0.69**	**−0.75**	**0.73**	**0.45**	**0.31**
P/L	**−0.44**	**−0.45**	**−0.43**	−0.08	**−0.53**	**−0.55**	**−0.59**	**−0.51**	**−0.43**	**0.30**	**0.32**	**0.25**
P	−0.14	**−0.31**	**−0.48**	**0.32**	−0.21	**−0.27**	**−0.71**	**−0.80**	**−0.82**	**0.76**	**0.54**	**0.40**
L	**0.57**	**0.42**	**0.23**	**0.44**	**0.61**	**0.59**	0.21	0.02	−0.12	**0.25**	−0.04	−0.08

GDD(DH): growing degree days to heading (°Cd); GDD(DM): growing degree days to maturity (°Cd); DH: days to heading (days); DM: days to maturity (days); GF: days of grain filling (days); GY: grain yield (t ha^−1^); HLW: hectoliter weight (kg/hl); TKW: thousand kernel weight (g/1000 kernels); NG: number of grains m^−2^ (units m^−2^); Prot (%): grain protein concentration (%); SDS: sodium dodecyl sulfate sedimentation (ml); W: dough strength (10^−4^ J); P/L: tenacity/extensibility ratio; P: tenacity (mm); L: extensibility (mm). Weather variables are followed by V (during the vegetative stage) and GF (during the grain filling stage). Tmin: minimum temperature (°C); TA: average temperature (°C); TM: maximum temperature (°C); PP: precipitation (mm); SR: global solar radiation (MJ m^−2^); ETo: potential evapotranspiration (mm).

**Table 5 plants-13-02929-t005:** Tukey’s mean separation tests among ten wheat varieties for plant traits, weather variables, and grain quality parameters for the trials in Lleida 2021 and 2022 (two water regimes and two sowing dates). The model used for crop traits and weather variables is the one including all the four fixed factors (year, sowing, water, and variety) and all their interactions, while quality traits were separated excluding the sowing factor (and all its interactions). Weather variables “V” and “GF” indicate those variables during the vegetative and grain filling stages, respectively. This table is sorted by ascending values of TM GF to highlight the pattern between phenology and weather exposure. Correlation coefficients are indicated for correlations performed using averages across all trials of each variety (10 points each correlation) between GY (“r with GY”) and the corresponding variable on the first column. Different superscript letters indicate significant differences at *p* < 0.05.

Tukey’s Mean Separation Among Varieties and Correlation Coefficients with GY											
Var. ID	7	4	1	3	10	6	9	2	8	5	r with GY
GDD(DH)	1096 ^g^	1089 ^g^	1121 ^f^	1155 ^e^	1157 ^e^	1164 ^e^	1310 ^d^	1368 ^c^	1406 ^b^	1432 ^a^	**−0.85**
GDD(DM)	1774 ^e^	1848 ^cde^	1831 ^e^	1836 ^de^	1866 ^cd^	1871 ^c^	2013 ^b^	2058 ^a^	2055 ^a^	2062 ^a^	**−0.75**
GF	36.7 ^bc^	40.4 ^a^	37.5 ^b^	35.5 ^c^	36.7 ^bc^	36.4 ^bc^	34.0 ^d^	32.6 ^e^	30.4 ^f^	29.3 ^f^	**0.90**
GY	7.9 ^abc^	8.3 ^a^	7.4 ^bcd^	7.0 ^cd^	8.0 ^ab^	8.4 ^a^	7.4 ^abcd^	6.8 ^d^	5.3 ^e^	5.3 ^e^	
TKW	31.3 ^bc^	33.3 ^b^	37.2 ^a^	28.9 ^de^	36.2 ^a^	35.6 ^a^	29.7 ^cd^	27.2 ^e^	28.6 ^de^	32.9 ^b^	0.47
NG	24.220 ^a^	24.184 ^a^	19.372 ^cd^	23.572 ^ab^	21.390 ^bc^	22.552 ^ab^	24.591 ^a^	24.265 ^a^	18.223 ^de^	16.365 ^e^	**0.70**
HLW	77.9 ^bc^	75.8 ^d^	78.5 ^b^	80.3 ^a^	80.1 ^a^	77.3 ^c^	71.1 ^f^	67.7 ^g^	72.4 ^e^	70.7 ^f^	0.56
Prot (%)	15.3 ^bc^	13.2 ^e^	14.8 ^c^	15.4 ^b^	14.8 ^c^	14.1 ^d^	15.5 ^b^	15.7 ^b^	16.6 ^a^	17.1 ^a^	**−0.91**
SDS	13.5 ^cd^	12.4 ^e^	13.9 ^bc^	14.3 ^ab^	13.9 ^abc^	12.1 ^e^	13.0 ^d^	12.3 ^e^	14.6 ^a^	13.3 ^cd^	−0.45
TM V	14.4 ^g^	14.4 ^g^	14.5 ^f^	14.6 ^e^	14.6 ^e^	14.7 ^e^	15.3 ^d^	15.5 ^c^	15.7 ^b^	15.8 ^a^	**−0.85**
SR V	1628.7 ^g^	1617.7 ^g^	1665.5 ^f^	1714.3 ^e^	1712.4 ^e^	1722.5 ^e^	1940.0 ^d^	2027.6 ^c^	2079.9 ^b^	2114.1 ^a^	**−0.85**
ETo V	222.7 ^g^	220.8 ^g^	229.0 ^f^	237.3 ^e^	236.9 ^e^	238.5 ^e^	276.8 ^d^	292.2 ^c^	301.8 ^b^	308.1 ^a^	**−0.85**
TM GF	26.6 ^g^	26.9 ^f^	27.1 ^f^	27.4 ^e^	27.6 ^d^	27.7 ^d^	29.3 ^c^	29.7 ^b^	29.8 ^ab^	30.0 ^a^	**−0.78**
SR GF	951.1 ^bc^	1048.4 ^a^	981.7 ^b^	936.1 ^cd^	975.9 ^b^	975.1 ^b^	911.0 ^d^	872.7 ^e^	819.4 ^f^	789.4 ^f^	**0.93**
ETo GF	172.2 ^c^	191.2 ^a^	179.4 ^b^	171.7 ^c^	179.5 ^b^	179.6 ^b^	172.6 ^c^	166.9 ^c^	157.1 ^d^	152.0 ^d^	**0.93**
ETo TOT	394.9 ^f^	412.0 ^cde^	408.4 ^e^	409.0 ^de^	416.4 ^cd^	418.1 ^c^	449.3 ^b^	459.1 ^a^	458.9 ^a^	460.1 ^a^	**−0.74**
W	286.4 ^ab^	159.6 ^ef^	276.1 ^abc^	331.5 ^a^	215.8 ^cdef^	192.3 ^def^	227.0 ^bcde^	149.9 ^f^	231.8 ^bcd^	182.3 ^def^	0.01
P/L	0.9 ^bc^	0.6 ^cd^	0.8 ^bc^	0.9 ^bc^	1.2 ^a^	0.9 ^b^	0.4 ^de^	0.3 ^e^	0.5 ^de^	0.4 ^de^	0.57
P	79.8 ^a^	51.8 ^de^	77.6 ^ab^	81.5 ^a^	76.1 ^ab^	66.4 ^bc^	53.9 ^de^	41.9 ^e^	58.0 ^cd^	48.9 ^de^	0.38
L	96.1 ^bcde^	84.9 ^de^	96.3 ^bcde^	94.4 ^cde^	65.0 ^e^	73.9 ^e^	130.0 ^ab^	137.9 ^a^	117.4 ^abcd^	124.3 ^abc^	**−0.67**

GDD(DH): growing degree days to heading (°Cd); GDD(DM): growing degree days to maturity (°Cd); GF: days of grain filling (days); GY: grain yield (t ha^−1^); HLW: hectoliter weight (kg/hl); TKW: thousand kernel weight (g/1000 kernels); NG: number of grains m^−2^ (units m^−2^); Prot (%): grain protein concentration (%); SDS: sodium dodecyl sulfate sedimentation (ml). Weather variables are followed by V (during the vegetative stage) and GF (during the grain filling stage). SR: solar radiation (MJ m^−2^); TM: maximum temperature (°C); ETo: potential evapotranspiration (mm); ETo TOT: total potential evapotranspiration from sowing to maturity; W: dough strength (10^−4^ J); P/L: tenacity/extensibility ratio; P: tenacity (mm); L: extensibility (mm). Bold font indicates significant Pearson correlation coefficients at *p* < 0.05.

**Table 6 plants-13-02929-t006:** Pearson correlation (corr) coefficients of plant traits with weather variables per trial are shown (using N = 10 varieties). Significant correlations are indicated in bold font (*p* < 0.05), and normal format indicates non-significant correlations.

Trials	ETo V					ETo GF					
	GY	Prot (%)	HLW	TKW	NG	GY	Prot (%)	HLW	TKW	NG	ETo TOT with GY
2021 Almacelles 1SD Irrigated	**−0.86**	**0.74**	**−0.92**	**−0.78**	−0.20	**0.67**	**−0.81**	0.45	0.49	0.31	**−0.75**
2021 Almacelles 2SD Irrigated	**−0.92**	**0.89**	**−0.83**	**−0.63**	**−0.70**	**0.86**	**−0.85**	0.54	0.44	**0.78**	**−0.85**
2021 Almacelles 1SD Rainfed	−0.61	0.43	**−0.90**	**−0.71**	0.02	**0.89**	**−0.85**	0.04	0.16	**0.74**	−0.35
2021 Almacelles 2SD Rainfed	−0.62	**0.74**	**−0.87**	**−0.66**	−0.33	**0.82**	**−0.88**	0.62	**0.70**	0.55	−0.38
2022 Sucs 1SD Irrigated	−0.29	0.19	**−0.74**	−0.35	0.15	0.46	−0.57	0.42	0.47	−0.12	−0.18
2022 Sucs 2SD Irrigated	**−0.80**	0.58	**−0.86**	−0.60	−0.28	**0.96**	**−0.93**	0.58	0.50	0.57	**−0.67**
2022 Sucs 1SD Rainfed	**−0.94**	**0.96**	**−0.71**	−0.16	**−0.75**	**0.96**	**−0.96**	0.50	0.09	**0.81**	**−0.88**
2022 Sucs 2SD Rainfed	**−0.95**	**0.86**	−0.61	**0.64**	**−0.96**	**0.96**	**−0.93**	0.52	−0.48	**0.90**	**−0.84**
*Irrigated 1SD	**−0.80**	0.58	**−0.86**	**−0.65**	−0.10	**0.76**	**−0.87**	0.55	**0.64**	0.14	**−0.71**
*Irrigated 2SD	**−0.90**	**0.82**	**−0.85**	−0.62	−0.58	**0.91**	**−0.96**	0.57	0.47	**0.74**	**−0.81**
*Rainfed 1SD	**−0.74**	**0.88**	**−0.83**	−0.52	−0.29	**0.95**	**−0.96**	0.31	0.27	**0.66**	−0.58
*Rainfed 2SD	**−0.75**	**0.84**	**−0.80**	−0.06	**−0.69**	**0.85**	**−0.97**	**0.66**	0.24	**0.70**	−0.60
**Trials**	**TM V**					**TM GF**					
	**GY**	**Prot** **(%)**	**HLW**	**TKW**	**NG**	**GY**	**Prot** **(%)**	**HLW**	**TKW**	**NG**	**TM TOT** **with GY**
2021 Almacelles 1SD Irrigated	**−0.85**	**0.74**	**−0.92**	**−0.78**	−0.19	**−0.79**	0.61	**−0.88**	**−0.71**	−0.19	**−0.74**
2021 Almacelles 2SD Irrigated	**−0.92**	**0.90**	**−0.82**	**−0.65**	**−0.68**	**−0.90**	**0.85**	**−0.84**	−0.61	**−0.70**	**−0.84**
2021 Almacelles 1SD Rainfed	−0.61	0.43	**−0.89**	**−0.71**	0.02	−0.51	0.34	**−0.91**	**−0.69**	0.10	−0.37
2021 Almacelles 2SD Rainfed	−0.61	**0.74**	**−0.89**	**−0.68**	−0.30	−0.54	**0.64**	**−0.88**	−0.62	−0.26	−0.40
2022 Sucs 1SD Irrigated	−0.30	0.20	**−0.73**	−0.34	0.14	−0.23	0.11	**−0.77**	−0.32	0.17	−0.18
2022 Sucs 2SD Irrigated	**−0.81**	0.59	**−0.86**	−0.60	−0.29	**−0.75**	0.52	**−0.85**	−0.59	−0.23	**−0.71**
2022 Sucs 1SD Rainfed	**−0.94**	**0.96**	**−0.71**	−0.16	**−0.76**	**−0.91**	**0.94**	**−0.74**	−0.21	**−0.69**	**−0.88**
2022 Sucs 2SD Rainfed	**−0.95**	**0.86**	−0.62	**0.65**	**−0.97**	**−0.77**	**0.67**	−0.52	0.51	**−0.81**	**−0.81**
*Irrigated 1SD	**−0.79**	0.58	**−0.86**	**−0.64**	−0.11	**−0.75**	0.49	**−0.85**	−0.59	−0.10	**−0.70**
*Irrigated 2SD	**−0.89**	**0.82**	**−0.87**	**−0.63**	−0.57	**−0.86**	**0.77**	**−0.85**	−0.61	−0.56	**−0.81**
*Rainfed 1SD	**−0.75**	**0.88**	**−0.82**	−0.52	−0.29	**−0.68**	**0.83**	**−0.86**	−0.54	−0.21	−0.59
*Rainfed 2SD	**−0.75**	**0.84**	**−0.81**	−0.05	**−0.68**	−0.62	**0.71**	**−0.78**	−0.04	−0.57	−0.60

ETo V (during the vegetative stage) and GF (during the grain filling stage): potential evapotranspiration (mm); TM V (during the vegetative stage) and GF (during the grain filling stage): maximum temperature (°C); GY: grain yield (t ha^−1^); Prot (%): grain protein concentration (%); HLW: hectoliter weight (kg/hl); TKW: thousand kernel weight (g/1000 kernels); NG: number of grains m^−2^ (units m^−2^); ETo TOT: potential evapotranspiration from sowing to maturity (mm); TM TOT: average maximum temperature from sowing to maturity (°C). * indicates correlations performed by averaging the two years of trials.

**Table 7 plants-13-02929-t007:** Pearson correlation (corr) coefficients of grain quality parameters with weather variables per trial are shown (using N = 10 varieties). Bold font indicates significant correlations (*p* < 0.05), while plain text indicates non-significant ones (*p* > 0.05).

Trials	ETo V				ETo GF				
	W	P/L	P	L	W	P/L	P	L	ETo TOT corr P/L
2021 Almacelles 1SD Irrigated	−0.30	**−0.84**	**−0.65**	**0.80**	−0.19	0.35	0.01	**−0.65**	**−0.84**
2021 Almacelles 2SD Irrigated	−0.09	−0.56	−0.44	0.58	−0.08	0.25	0.10	−0.27	**−0.67**
2021 Almacelles 1SD Rainfed	−0.42	**−0.73**	**−0.75**	**0.66**	−0.40	−0.06	−0.23	−0.15	**−0.81**
2021 Almacelles 2SD Rainfed	−0.42	−0.59	−0.60	0.56	−0.03	0.36	0.21	−0.47	−0.63
2022 Sucs 1SD Irrigated	−0.16	**−0.65**	−0.45	0.61	−0.11	0.15	−0.01	−0.29	**−0.78**
2022 Sucs 2SD Irrigated	−0.45	**−0.78**	**−0.78**	**0.66**	0.07	0.58	0.40	−0.57	**−0.78**
2022 Sucs 1SD Rainfed	−0.22	−0.58	−0.52	**0.76**	0.03	0.43	0.35	**−0.72**	−0.63
2022 Sucs 2SD Rainfed	−0.44	−0.46	−0.56	**0.64**	0.15	0.58	0.53	**−0.79**	−0.30
*Irrigated 1SD	−0.30	**−0.78**	−0.60	**0.80**	−0.15	0.24	0.03	−0.58	**−0.88**
*Irrigated 2SD	−0.26	**−0.73**	**−0.64**	**0.68**	−0.04	0.48	0.26	−0.48	**−0.78**
*Rainfed 1SD	−0.38	**−0.70**	**−0.70**	**0.79**	−0.09	0.29	0.22	−0.52	**−0.77**
*Rainfed 2SD	−0.48	−0.61	**−0.68**	**0.70**	0.13	0.57	0.44	**−0.72**	−0.58
**Trials**	**TM V**				**TM GF**				
	**W**	**P/L**	**P**	**L**	**W**	**P/L**	**P**	**L**	**TM TOT corr P/L**
2021 Almacelles 1SD Irrigated	−0.30	**−0.83**	**−0.65**	**0.79**	−0.38	**−0.80**	**−0.70**	**0.68**	**−0.82**
2021 Almacelles 2SD Irrigated	−0.07	−0.55	−0.43	0.58	−0.14	−0.56	−0.47	0.54	**−0.66**
2021 Almacelles 1SD Rainfed	−0.42	**−0.72**	**−0.74**	**0.64**	−0.48	**−0.76**	**−0.81**	**0.67**	**−0.80**
2021 Almacelles 2SD Rainfed	−0.43	−0.62	−0.62	0.60	−0.53	−0.60	**−0.67**	0.55	**−0.66**
2022 Sucs 1SD Irrigated	−0.16	**−0.65**	−0.45	0.60	−0.18	**−0.72**	−0.51	**0.67**	**−0.78**
2022 Sucs 2SD Irrigated	−0.45	**−0.78**	**−0.78**	**0.66**	−0.46	**−0.76**	**−0.79**	**0.66**	**−0.79**
2022 Sucs 1SD Rainfed	−0.23	−0.57	−0.52	**0.76**	−0.23	−0.58	−0.52	**0.74**	**−0.63**
2022 Sucs 2SD Rainfed	−0.45	−0.48	−0.57	**0.65**	−0.43	−0.20	−0.35	0.47	−0.28
*Irrigated 1SD	−0.30	**−0.78**	−0.61	**0.80**	−0.36	**−0.82**	**−0.67**	**0.77**	**−0.88**
*Irrigated 2SD	−0.27	**−0.75**	**−0.65**	**0.69**	−0.30	**−0.72**	**−0.66**	**0.65**	**−0.79**
*Rainfed 1SD	−0.39	**−0.69**	**−0.70**	**0.78**	−0.42	**−0.72**	**−0.73**	**0.78**	**−0.77**
*Rainfed 2SD	−0.49	−0.63	**−0.69**	**0.72**	−0.56	−0.53	**−0.67**	0.61	−0.59

ETo V (during the vegetative stage) and GF (during the grain filling stage): potential evapotranspiration (mm); TM V (during the vegetative stage) and GF (during the grain filling stage): maximum temperature (°C); W: dough strength (10^−4^ J); P/L: tenacity/extensibility ratio; P: tenacity (mm); L: extensibility (mm); ETo TOT: potential evapotranspiration from sowing to maturity (mm); TM TOT: average maximum temperature from sowing to maturity (°C). * indicates correlations performed by averaging the two years of trials.

**Table 8 plants-13-02929-t008:** Pearson correlation coefficients of hectoliter weight, along with average values (AVG), with grain quality parameters per trial, and averaging all trials, are shown (using N = 10 varieties). Bold font indicates significant correlations (*p* < 0.05), while plain indicates text non-significant ones (*p* > 0.05).

Trials	HLW				AVG HLW
	W	P/L	P	L	
2021 Almacelles 1SD Irrigated	0.51	**0.94**	**0.81**	**−0.77**	77.9
2021 Almacelles 2SD Irrigated	0.33	**0.90**	**0.79**	**−0.83**	73.9
2021 Almacelles 1SD Rainfed	0.61	**0.92**	**0.93**	**−0.77**	80.5
2021 Almacelles 2SD Rainfed	0.61	**0.85**	**0.82**	**−0.81**	77.7
2022 Sucs 1SD Irrigated	0.23	**0.72**	0.59	**−0.75**	75.7
2022 Sucs 2SD Irrigated	0.58	**0.85**	**0.92**	**−0.74**	73.2
2022 Sucs 1SD Rainfed	0.43	**0.67**	**0.65**	**−0.76**	70.5
2022 Sucs 2SD Rainfed	0.61	0.57	**0.75**	−0.46	72.3
*Irrigated 1SD	0.54	**0.89**	**0.78**	**−0.81**	76.8
*Irrigated 2SD	0.52	**0.92**	**0.90**	**−0.81**	73.5
*Rainfed 1SD	0.62	**0.85**	**0.87**	**−0.85**	75.5
*Rainfed 2SD	**0.65**	**0.82**	**0.88**	**−0.76**	75.0

HLW: hectoliter weight (kg/hl); W: dough strength (10^−4^ J); P/L: tenacity/extensibility ratio; P: tenacity (mm); L: extensibility (mm). * indicates correlations performed by averaging the two years of trials.

## Data Availability

The original data presented in this study are openly available at https://figshare.com/, at https://doi.org/10.6084/m9.figshare.26334766.

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
