# Peer review of "Phenological Adaptation of Wheat Varieties to Rising Temperatures: Implications for Yield Components and Grain Quality"

_plants, 2024, doi:10.3390/plants13202929_

Round 1
Reviewer 1 Report
Comments and Suggestions for Authors
This paper presents an account of a substantial body of work examining the relation between weather variables and the yield of wheat. A major strength of the work is the comparison of effects on both yield (offtake mass per unit area) and grain quality (protein content, etc.). Few studies examine both. Further strengths include the range of conditions over which factors were assessed (sites, agronomic treatments and years) and the division of phenology into vegetative and grain filling stages. Studies of this type are important for assessing current limitations to yield and quality but also future effects of changing weather and climate.
However, there are serious concerns about the data presented on climate (temperature, evapotranspiration, solar irradiance) and the method of linking climatic factors to yield and quality. There seems to be apparent inconsistencies between climatic data used in different parts of the Results. As it stands, this reviewer considers the original data as very suitable in principle as the basis of a scientific publication but the presentation of climate and the analysis linking climatic variables to effect is not up to standard. The recommendation is ‘reject’ as the paper stands, but consider improving the presentation and analysis of climatic data with a view to a new submission to this or another journal. The authors might also wish to comment on the potential effects of the experimental plots being narrow (1.2 m) and so prone to ‘edge effects’
Main concerns are as follows.
1. Clarity on measurement of weather and climate
Greater clarity is needed as to how the data on weather and climate in Table 2 were calculated. This is important because weather is such a crucial part of the overall analysis. Although a link is given to a Catalonia met network, the methods of measurement need to be stated here. For example, maximum and minimum air temperatures were presumably the mean of the daily max and min over each month, but how was the monthly average temperature calculated? It is also important to state the devices used to measure temperature at the two met sites.
Similarly, the methods of measuring the other variables need to be stated. For example, was solar radiation measured by a Kipp and Zonen or similar, and are the values in Table 2 the sum of all daily values in a month? But more important is the measurement of evapotranspiration (ETo) – section 2.1 describes the method of estimating irrigation needs through the Penman-Monteith approach modified by a crop coefficient to get crop evaporation (ETc) , but how was ETo measured to give the data in Table 2 – from evaporation pans at a met site or from calculations?.
Further comments: why in section 2.1 (first paragraph) was rainfall and evapotranspiration given ‘across both locations’ when only one was used; and note that annual rainfall total is omitted from the text for the first year.
Also, how were the weather variables measure or estimated for the 26 additional trials conducted over various place and years?
2. Weather variables in Table 2 highly correlated
There is more emphasis on ETo in the analysis (e.g. Fig. 1) but in Table 2, ETo, temperature and solar radiation are very highly correlated at both site-years. The correlation is slightly lower between ETo and maximum temperature than between ETo and solar radiation (SR). Correlations between such factors are to be expected to a degree, but in many places ETo is affected by additional variables such as wind speed and boundary layer. Is there any explanation for such high correlation among the monthly averages and totals? Please also see point 6.
3. Outdated approach to linking weather to yield.
Since the 1960s and 1970s, crop physiologists, crop modellers and agronomists have been interpreting the effects of weather and climate on crops through variables such as (i) intercepted or absorbed solar radiation rather than total incoming radiation; (ii) thermal time (temperature in relation to base and optimal temperatures accumulated over a period of growth), and (iii) water use efficiency and related metrics, e.g. crop dry matter or yield in relation to water used by the crop. Yet here crop data are related to raw met parameters with the added uncertainty that brings to the analysis due to the absence of physiological links between yield and weather. It would also really help to define high-temperature limitations if some attempt were made to estimate the optimum temperature for phenology, i.e. the temperature at which developmental phases are shortest, but above which development and related factors are limited.
4. Crop phenology should be described earlier
Fig. 2 is a good representation of the overall trends in crop phenology and some weather variables but which site(s) does the figure refer to? To help the reader’s understanding of phenology, this figure should be ideally placed near the beginning of the Results, since it sets the scene for much of the analysis. It could also include cumulative ETo or the cumulative difference between ETo and rainfall, and also cumulative solar radiation. The horizontal (date) axis is hard to follow and should be revised.
5. Analysis in Fig 1 is inconsistent
Analysis of the type in Fig. 1 between crop yield and weather variables is essential for visual demonstration of differences between sites and treatments. But if in Table 2, temperature and ETo are highly correlated, why in Fig. 1 are they so different. In Fig. 2, max temperature is higher at Sucs 2022 than at Almacelles 21, consistent with differences in Table 2 in May and June; but ETo is the other way round – lower at Sucs 22 than at Almacelles 21, which is not consistent with the correlation between temperature and ETo and with the data in Table 2.
6. Applied Nitrogen and protein content
The different effects of weather on yield and protein content is an important finding, but protein content depends on other factors than weather. It is stated in the methods that trials were optimally fertilised but does this mean they were all given the same amount of nitrogen or did that the amount vary depending on expected yield? Values for N fertiliser should be stated.
7. Emphasis on statistical values in Tables
Several of the tables consist of correlation coefficients and other statistical output but agronomists and crop physiologists would like to see actual data, for example of weather variables during the vegetative and grain filling phases. Can the actual data be introduced to the Tables and some of the statistical results moved to supplementary data?
Comments on the Quality of English Language
Minor checks and amendments only.
Author Response
REVIEWER 1
Main concerns are as follows.
- Clarity on measurement of weather and climate
Greater clarity is needed as to how the data on weather and climate in Table 2 were calculated. This is important because weather is such a crucial part of the overall analysis. Although a link is given to a Catalonia met network, the methods of measurement need to be stated here. For example, maximum and minimum air temperatures were presumably the mean of the daily max and min over each month, but how was the monthly average temperature calculated? It is also important to state the devices used to measure temperature at the two met sites.
Similarly, the methods of measuring the other variables need to be stated. For example, was solar radiation measured by a Kipp and Zonen or similar, and are the values in Table 2 the sum of all daily values in a month? But more important is the measurement of evapotranspiration (ETo) – section 2.1 describes the method of estimating irrigation needs through the Penman-Monteith approach modified by a crop coefficient to get crop evaporation (ETc) , but how was ETo measured to give the data in Table 2 – from evaporation pans at a met site or from calculations?.
Further comments: why in section 2.1 (first paragraph) was rainfall and evapotranspiration given ‘across both locations’ when only one was used; and note that annual rainfall total is omitted from the text for the first year.
Also, how were the weather variables measure or estimated for the 26 additional trials conducted over various place and years?
Response 1:
- A) Regarding the first comment on ‘edge effects’ in 1.2m wide plots, we acknowledge that the outer rows in an 8-row wheat plot are subject to edge effects. However, the six inner rows, which are not affected by these edge effects, outnumber the two outer rows. Moreover, all plots and treatments are exposed to the same edge effects on the outer rows. While edge effects might be a concern when calculating yield in 2, 4, or 6-row plots, we may assume that with 8 rows, the impact of edge effects is negligible.
- B) Regarding the details of temperature measurement calculations see lines 221-227:
Temperature along with other weather parameters, was recorded on an hourly basis.
Mean Maximum Temperature: The average of the highest temperatures recorded each day, calculated over each month.
Mean Minimum Temperature: The average of the lowest temperatures recorded each day, calculated over each month.
Average Temperature: the average of all the temperatures recorded over each month.
C)Regarding the request for clarity on measurement of weather and climate:
Our meteorological station has the following devices:
- Rainfall was measured using an ECRN-50 rain gauge with 0.25 mm resolution, connected to a EM50 data logger (Decagon Devices)
- Solar radiation was measured with a QSO-S PAR Photon Flux sensor measures the Photosynthetic Photon Flux (PPF) in µmol m-2 s-1 from a field of view of 180 degrees (Decagon Devices)
- Air temperature, humidity and air pressure using the Decagon VP-4 system (Decagon Devices)
- Wind speed measured with Davis Cup4 Anemometer (Decagon Devices)
This information has been added to the Materials and Methods section see lines 211-221.
All meteorological data for he additional 26 field trials was collected from official meteorological stations, equipped with the specified devices available at: https://www.meteo.cat/wpweb/divulgacio/equipaments-meteorologics/estacions-meteorologiques-automatiques/xarxa-destacions-meteorologiques-automatiques-xema/els-sensors/
- D) Evapotranspiration is a calculation: ETc was computed as the product of the reference potential evapotranspiration (ETo), calculated using the Penman–Monteith method [29] and the crop coefficients (Kc) approach, derived from FAO-56, and adjusted based on prevailing weather conditions before being utilized for ETc computations (explained in lines 146-153).
Regarding the calculation of Evapotranspiration in Table 2: ETo was calculated as the sum of the daily crop evapotranspiration (Etc) during each month. Moreover, the cumulative evapotranspiration during the vegetative and grain filling stages for each wheat variety (in Table 5) was calculated as explained under the M&M subsection “2.3. Weather Parameters” lines 230-243.
For the cumulative potential Evapotranspiration of each variety we have used the following assumption: In calculating the cumulative potential evapotranspiration for each variety, it was assumed that the main differences between varieties in terms of exposure to temperature, precipitation, and evapotranspiration are primarily due to the duration and extent of the crop cycle. The intrinsic differences in evaporation and transpiration due to biomass variations in different wheat varieties, were considered negligible when compared to the overall water balance over time throughout the different crop cycles of each variety. This has been demonstrated and supported by Gomez-Candón et al. 2023, who found that the actual evapotranspiration of each variety is closely correlated with the reference evapotranspiration. (Gómez-Candón, D.; Bellvert, J.; Pelechá, A.; Lopes, M.S. A Remote Sensing Approach for Assessing Daily Cumulative Evapotranspiration Integral in Wheat Genotype Screening for Drought Adaptation. Plants 2023, 12, 3871. https://doi.org/10.3390/plants12223871). See new text introduced under “2.3. Weather Parameters” M&M section in lines 243-252.
- E) Regarding the point on ‘further comments: why in section 2.1 (first paragraph) was rainfall and evapotranspiration given ‘across both locations’ when only one was used; and note that annual rainfall total is omitted from the text for the first year.’
Author Response: This was an oversight, and ‘across both locations’ has been removed from the text. The average total annual rainfall (280.7 mm) has now been added to the text in line 124.
- F) Regarding the point: ‘Also, how were the weather variables measure or estimated for the 26 additional trials conducted over various place and years?’
Author Response: We utilized data from the official network of meteorological stations across Catalonia. http://www.ruralcat.net/web/guest/agrometeo. See line 215-221.
- Weather variables in Table 2 highly correlated
There is more emphasis on ETo in the analysis (e.g. Fig. 1) but in Table 2, ETo, temperature and solar radiation are very highly correlated at both site-years. The correlation is slightly lower between ETo and maximum temperature than between ETo and solar radiation (SR). Correlations between such factors are to be expected to a degree, but in many places ETo is affected by additional variables such as wind speed and boundary layer. Is there any explanation for such high correlation among the monthly averages and totals? Please also see point 6.
Response 2: Thank you for these insightful observations. The high correlation among the weather variable values in Table 2 can be attributed to the variations captured, which are related to the monthly fluctuations from January through June. Although weather patterns can be erratic in terms of probability, it is common in this region to observe a progressive increase in temperature and a decrease in precipitation from January to June, each year. This leads to the observation of high correlations among weather parameters between the years.
Below the Reviewer can find wind speed data (not included in the manuscript, as the current tables already contain a substantial amount of information). This additional table will clarify that the correlations in Table 2 are specific to the months listed. Wind speed is highly variable—strong winds can occur in January, May, or June—resulting in lower correlations overall.
- Outdated approach to linking weather to yield.
Since the 1960s and 1970s, crop physiologists, crop modellers and agronomists have been interpreting the effects of weather and climate on crops through variables such as (i) intercepted or absorbed solar radiation rather than total incoming radiation; (ii) thermal time (temperature in relation to base and optimal temperatures accumulated over a period of growth), and (iii) water use efficiency and related metrics, e.g. crop dry matter or yield in relation to water used by the crop. Yet here crop data are related to raw met parameters with the added uncertainty that brings to the analysis due to the absence of physiological links between yield and weather. It would also really help to define high-temperature limitations if some attempt were made to estimate the optimum temperature for phenology, i.e. the temperature at which developmental phases are shortest, but above which development and related factors are limited.
Response 3: We thank the reviewer for these valuable insights. Indeed, intercepted solar radiation, thermal time, and water use efficiency are critical aspects of crop physiology, agronomy, and modelling that have been extensively studied by many distinguished researchers worldwide. First, we have now calculated growing degree days and is shown in all tables.
Secondly, the approach used in our study is different from what has been published, and considered how the weather is affecting each variety at specific developmental stages; we specifically considered the cumulative effects of temperature, water availability, and evapotranspiration during the two crucial developmental phases of wheat: vegetative growth and grain filling. This type of analysis has not been previously developed in the studies from the 1960s and 1970s. Furthermore, the significance of monthly weather parameters has been largely overlooked in earlier research. Peaks in temperature or periods of drought during critical developmental months have a much greater impact on final yield than the average conditions over the entire crop cycle, where the effects of specific months tend to be diluted. We hypothesized that the observed links between weather exposure during these two critical developmental phases were essential for yield formation, and this was demonstrated in our study. The concept of focusing on monthly temperatures arises from field observations indicating that the lack of water, precipitation, or optimal temperature during the stem elongation phase is particularly critical for yield outcomes—something well recognized by farmers. This phase is more sensitive than, for example, the tillering stage, where a lack of precipitation would not have been as damaging. When considering total rainfall or average temperature across the entire crop cycle, this effect may be diluted. By dilution, we mean that cooler, high-precipitation conditions in February (compared to the average long-term conditions) and warmer, drier conditions in April may offset each other, resulting in a neutral overall effect when averages are considered.
Regarding the last question: “It would also really help to define high-temperature limitations if some attempts were made to estimate the optimum temperature for phenology, i.e., the temperature at which developmental phases are shortest, but above which development and related factors are limited.” Our analysis found a linear and negative correlation between heading date/growing degree days and temperature, suggesting that the earliest varieties were already exposed to critical temperatures (please see graph below). Specifically, the earliest varieties experienced an average maximum temperature of 14.4°C during the vegetative phase, and 26.6°C during the grain filling phase (means of all years and experimental conditions)-see lines 444-449. Beyond these temperatures, yield begins to decrease in a linear manner, as shown in the graph below. Moreover, as shown previously by other researchers (Hellemans et al. 2018; Saint Pierre 2008), above 30oC dough strength starts to decrease, however the effect on phenology with visible consequences on yield may start earlier. The challenge remains that we do not have data on even earlier varieties than those included in this study to identify the exact inflection point. However, it is important to note that in this region, very early varieties are at risk of late frosts, which has led to their discontinuation. A sentence was added in the results sections, in the new version “Furthermore, the earliest varieties experienced an average maximum temperature of 14.4°C during the vegetative phase and 26.6°C during the grain filling phase. Yields consistently declined at temperatures above these thresholds, as demonstrated in Table 4, which includes data from the earliest variety (4) to the latest variety (5) and all varieties in between”; and in the discussion section. See lines 434-439.
- Crop phenology should be described earlier
Fig. 2 is a good representation of the overall trends in crop phenology and some weather variables but which site(s) does the figure refer to? To help the reader’s understanding of phenology, this figure should be ideally placed near the beginning of the Results, since it sets the scene for much of the analysis. It could also include cumulative ETo or the cumulative difference between ETo and rainfall, and also cumulative solar radiation. The horizontal (date) axis is hard to follow and should be revised.
Response 4: Figure 2 has been moved to the first subsection of the results section and is now referred to as Figure 1 (page 7). We have improved the plot by adding potential evapotranspiration and global solar radiation, allowing for a graphical representation of exposure to all the analyzed meteorological variables across all varieties. For the sake of conciseness, we opted to present the data in each year and location (see new Figure 1). Varietal diversity is crucial, particularly in these environments, for determining weather exposure of the crop during grain filling. Figure 2 (now Figure 1) clearly demonstrates that early-cycle varieties experienced more favorable weather conditions, with lower temperature exposure and higher levels of rainfall, potential evapotranspiration, and global solar radiation compared to late-cycle varieties. We have now moved this figure up the results section, see page 7.
- Analysis in Fig 1 is inconsistent
Analysis of the type in Fig. 1 between crop yield and weather variables is essential for visual demonstration of differences between sites and treatments. But if in Table 2, temperature and ETo are highly correlated, why in Fig. 1 are they so different. In Fig. 2, max temperature is higher at Sucs 2022 than at Almacelles 21, consistent with differences in Table 2 in May and June; but ETo is the other way round – lower at Sucs 22 than at Almacelles 21, which is not consistent with the correlation between temperature and ETo and with the data in Table 2.
Response 5: Thank you for highlighting these differences. While the calculations are correct, the explanations of how each one is derived may not be entirely clear and are explained below:
The discrepancies between Figure 1 (now Figure 2) and Table 2 are primarily due to the different calculation methods used and purposes. In Table 2 raw weather data is presented, Eto was calculated as a monthly average throughout the crop cycle, based on raw weather data averaged per month. In contrast, Figure 1 (now Figure 2) shows the average Eto during the vegetative and grain filling stages, calculated for each wheat variety depending on their phenology (date of heading, maturity and grain filling duration). This calculation is detailed in the Materials and Methods section on Page 5, Lines 230-239: ‘averages and cumulative values were separately computed for the vegetative and grain filling stages to assess the relative exposure to these weather variables during the two different phases of plant cycle in each wheat variety. This process involved the following methods: averaged values (for temperature variables) were derived, for the vegetative stage, by dividing the sum of daily measurements from sowing to heading date by the number of days in this period (days to heading); for the grain filling stage, the sum of daily measurements from heading to maturity date was divided by the number of days in this period (grain filling duration). Cumulative values (for PP, SR, ETo and TOT W) were calculated by summing the daily measurements for each specified period. Consequently, all these newly calculated variables were designated by adding “V” for those calculated during the vegetative stage and “GF” for those calculated during the grain filling stage’
- Applied Nitrogen and protein content
The different effects of weather on yield and protein content is an important finding, but protein content depends on other factors than weather. It is stated in the methods that trials were optimally fertilised but does this mean they were all given the same amount of nitrogen or did that the amount vary depending on expected yield? Values for N fertiliser should be stated.
Response 6: The trials were conducted in a region with high baseline nitrogen levels, this is a major problem in the Catalonia Region due to extremely high animal production. We calculated the N fertilization required to the highest yields obtained in our experimental station (can reach 13-15 T/Ha). In the 2021 crop cycle, no nitrogen fertilizer was applied because the soil already contained over 250 units of nitrogen. However, in the 2022 crop cycle, 27 units of nitrogen were applied before sowing in the form of Diammonium Phosphate (18-46). This information has been included in the Methods and Materials section. See lines 128-132 ‘During the 2021 crop cycle, no nitrogen fertilizer was applied due to the soil's nitrogen content exceeding 250 units. However, in the 2022 crop cycle, 27 units of N were applied before sowing using Di-ammonium Phosphate (18-46). ’
- Emphasis on statistical values in Tables
Several of the tables consist of correlation coefficients and other statistical output but agronomists and crop physiologists would like to see actual data, for example of weather variables during the vegetative and grain filling phases. Can the actual data be introduced to the Tables and some of the statistical results moved to supplementary data?
Response 7: Agreed. We have now moved Tables 3, 4, and 7, which contain the ANOVA results, to the supplementary material (see supplementary materials). The main text now includes the tables with the actual data and the correlation analysis results.

Reviewer 2 Report
Comments and Suggestions for Authors
There are some problems in the manuscript listed as follow:
1 Line 77-79, Additionally, it assessed how variations in wheat phenology affect grain yield components and grain quality, with the aim of identifying opportunities for selecting wheat varieties that are better adapted to climate. The English tense is inconsistency in this sentence.
2 Line 96-97, ‘Ten distinct commercial varieties were cultivated’, but no detail information to describe them.
3 Line 200, the unit of the Tmin should be supplied in the table 2.
4 Line 221, the unit presentation of the P (%) is different from the other factors in the table 3. It is easy to be confused with the protein content (P %) in line 144 and tenacity (P in mm) in line 149.
5 Line 423, the formation of the reference should be satisfied with the requirements of the journal, such as ‘40. A. Blum, Field Crops Res 112 (2009) 119–123’.
Comments on the Quality of English Language
The writting of this manu should be improved for the English tense.
Author Response
This is the correct response to the questions and comments from Reviewer 2. Previously, the responses provided here were intended for Reviewer 1 even if the doc file below has the right answers to Rev. 2. We apologize for the confusion. Please find the responses to Reviewer 2 below.
Comments and Suggestions for Authors
There are some problems in the manuscript listed as follow:
Comment 1: Line 77-79, Additionally, it assessed how variations in wheat phenology affect grain yield components and grain quality, with the aim of identifying opportunities for selecting wheat varieties that are better adapted to climate. The English tense is inconsistency in this sentence.
Response 1: The last part of the sentence was changed for clarity: ‘Finally, the study evaluated whether phenology influences grain quality parameters. The outcomes of this research will help refining selection criteria in breeding programs aimed at improving wheat adaptation to climate change.’ The objective was to determine if phenology had any effects on grain quality parameters, which was something that was not addressed previously and has implications for the selection criteria in breeding programs aiming to increase adaptation to climate change. See lines 108-111.
Comment 2: Line 96-97, ‘Ten distinct commercial varieties were cultivated’, but no detail information to describe them.
Response 2: To avoid potential conflicts and competition among private seed companies, we have decided not to disclose this information publicly. However, we have provided the names of the varieties below for the Reviewer's reference:
|
Siskin |
KWS |
|
CHAMBO |
Limagrain |
|
ALGORITMO RGT |
RAGT Seeds |
|
MARCOPOLO |
RAGT Seeds |
|
SOBERBIO |
Caussade |
|
RGT Reform |
RAGT Seeds |
|
NOGAL |
Florimond Desprez |
|
Hondia |
Florimond Desprez |
|
BOLOGNA |
Semillas Batle |
|
Benchmark |
Lemaire Deffontaines |
Comment 3: Line 200, the unit of the Tmin should be supplied in the table 2.
Response 3: Done. See Table 2 on page 8.
Comment 4: Line 221, the unit presentation of the P (%) is different from the other factors in the table 3. It is easy to be confused with the protein content (P %) in line 144 and tenacity (P in mm) in line 149.
Response 4: We have replaced P (%) with Prot (%) across the manuscript to avoid this confusion.
Comment 5: Line 423, the formation of the reference should be satisfied with the requirements of the journal, such as ‘40. A. Blum, Field Crops Res 112 (2009) 119–123’.
Response 5: Done. Reference 47 – Page 24.
Comments on the Quality of English Language
The writting of this manu should be improved for the English tense.
Response 6: We have revised the English throughout the manuscript to improve clarity.
Submission Date
18 July 2024
Date of this review
28 Jul 2024 16:25:12

Reviewer 3 Report
Comments and Suggestions for Authors
Dear Authors, I have reviewed the manuscript and provide my comments below:
This manuscript investigated the combined effects of late sowing, water limitation and inter-annual weather changes on grain yield and quality for ten wheat varieties. It was found that delayed sowing and water deficit significantly reduced yield, while changes in grain quality had an effect only under rainy conditions. The topic is, I think, very topical, because climate change is a very important factor that requires us to cultivate new technologies and to look at our current knowledge from a new perspective. The subject is a good one, but there are a couple of things that I think are worth amending:
The structure of the Introduction chapter is good, but there are too many references to old publications, in contrast to the paucity of data and results from the last 5 years, although there are some new results on the topic. Therefore, I suggest rewriting and expanding the chapter.
In parallel, it is proposed to rewrite the Discussion chapter and to explore deeper connections between publications and own results. The structure of the Introduction and Discussion chapters should be similar.
The research design should also be better highlighted in the Introduction chapter of the manuscript and in the Abstract.
Author Response
REVIEWER 3
Comments and Suggestions for Authors
Dear Authors, I have reviewed the manuscript and provide my comments below:
This manuscript investigated the combined effects of late sowing, water limitation and inter-annual weather changes on grain yield and quality for ten wheat varieties. It was found that delayed sowing and water deficit significantly reduced yield, while changes in grain quality had an effect only under rainy conditions. The topic is, I think, very topical, because climate change is a very important factor that requires us to cultivate new technologies and to look at our current knowledge from a new perspective. The subject is a good one, but there are a couple of things that I think are worth amending:
The structure of the Introduction chapter is good, but there are too many references to old publications, in contrast to the paucity of data and results from the last 5 years, although there are some new results on the topic. Therefore, I suggest rewriting and expanding the chapter. In parallel, it is proposed to rewrite the Discussion chapter and to explore deeper connections between publications and own results. The structure of the Introduction and Discussion chapters should be similar.
Response 1: The introduction section has been restructured to align with the organization of the discussion section. It now first addresses the environmental effects, followed by the available genetic strategies to cope with high temperature and drought stress (see lines 33-111).
In the new Discussion, we have newly discussed the results presented by Liu et al. 2022 where 4,096 hypothetical genotypes were constructed to understand adaptation to late sowing. In this case, longer juvenile phases were better adapted, with greater grain filling rates. While these results may appear to contradict those presented here with field experiments, they highlight that adaptation strategies can be specifically tailored to suit different environments and cropping systems. See lines 706-711.
The research design should also be better highlighted in the Introduction chapter of the manuscript and in the Abstract.
Response 3: A sentence has been added to the abstract and to the introduction section about the research design. See lines 10-13 in the abstract and 97-101 for the introduction section.
Submission Date
18 July 2024
Date of this review
21 Aug 2024 08:53:09

Round 2
Reviewer 1 Report
Comments and Suggestions for Authors
This reviewer appreciates the changes made by the authors in response to comments on Version 1. Many of the original objections have been cleared by new data. Below, the reviewer comments on each of the author’s Responses (1 to 7) detailed in their coverletter. Most responses have been classed as ‘acceptable’ but further work is recommended on several, especially Response 3 (and see paragraph on evapotranspiration below). In addition, the new material introduces several inconsistencies such as that between the measurements of photon flux and that presented in the Table as total solar radiation.
In general, then, the paper is improved, but the reviewer is perplexed by the continued emphasis on evapotranspiration in the Abstract and elsewhere. I suspect most readers with any background in crop-weather relations will be similarly perplexed. The problem is that evapotranspiration is a result of both weather conditions (primarily solar income, wind speed and temperature) and some characteristics of the crop. It is not a primary driver of crop production. In many parts of the world, high evapotranspiration is assessed as a negative influence on crop growth – for each unit of carbon assimilated by the plant, more water is needed for transpiration and surface evaporation. Here the correlation is probably a false one, the primary driver being solar radiation intercepted by the crop over the periods of vegetative and grain growth. The authors need to discuss these conflicts more fully.
Reviewer’s comment on Authors’ Responses:
1A. The edge effects in an 8-row plot are not negligible. The authors should state clearly in the Discussion that yields are from small plots and – though the comparative effects across treatments might be acceptable – the actual yields would not be representative of those from full-sized fields.
IC. The new information on other met measurements is acceptable with the following exception – if photon flux is the primary measure of incoming solar, why are data in Table 2 presented in MJ, which implies total solar radiation?
IB, 1D, IE and IF. Amendments and responses acceptable.
Response 2. This is a plausible explanation.
Response 3. The explanation in the paragraph beginning “Secondly, the approach ..” does not stand scrutiny. There have been many previous analyses of the importance of weather in specific developmental phases to final outcomes. The authors should not claim that their approach here is radically new.
The response to “Regarding the last question….” The arguments given do not cover all possibilities. Yes, there is a strong relation between degree days and maximum temperature, but this is to be expected since the maximum will be highly correlated with the mean. The question unresolved is why increasing in maximum temperature is negatively correlated with grain yield. The analysis so far has not distinguished the following possibilities:
· That yield is reduced by a direct effect of high temperature inhibiting grain filling or related processes – which is what the authors seem to prefer.
· That increasing temperature shortens the duration of grain filling (which is a commonly observed effect) and by doing so reduces the amount of solar radiation that the crops capture during this phase and therefore the amount of CO2 taken in and dry matter produced.
· That the genotypes have inherently different responses to high temperature.
· That other factors such as crop cover (affecting solar radiation interception) and capacity for translocation vegetative dry matter to grain also differ among varieties.
Data in the new Table 4 suggest yield is more strongly correlated with the duration of grain filling than with maximum temperature, which is consistent with the second option above – that high temperature is speeding up development and simply reducing the period over which the crop absorbs solar radiation. The paper would benefit from a discussion on these possibilities and a statement that a final resolution is not possible with the data gathered. The new section on threshold temperature in the first paragraph of the Discussion needs to be re-thought.
Response 4. This change in placement greatly improves the flow of the Results. A very useful figure.
Response 5. A plausible explanation but the apparent discrepancy still needs explaining in full.
Response 6. An acceptable explanation but what are ‘units’ of nitrogen – N in soil or applied as fertilizer is usually quantified in kg/ha or similar? Please clarify. And while the reason for the high existing soil N is explained in the response, it needs explaining in the Methods, since it is very unusual for a high-N requiring crops such as wheat to have no N applied.
Response 7. Acceptable.
Further comments
Table 3. Please check all abbreviations and units e.g. degree days is usually given in units of oCd not oC.
Table 4 – difficult to tell which heading refers to a given column – e.g. is it TM V or VETo; no need to repeat as in ‘days to heading (days)’; define GDD and provide units.
Fig. 1. Giving dates on the horizontal axis as month/day/year every four days or so – and without a line drawn for the ais - makes it very difficult to work out the timings of the varieties. The axis should be simply numbered (e.g. by day or year) with tick marks every ten days or thereabout and additional pointers to indicate months.
Fig. 2 and elsewhere – units of mass – ton is not generally used in the SI systems, why not use t ha-1.
Author Response
Reviewer’s comment on Authors’ Responses:
1A. The edge effects in an 8-row plot are not negligible. The authors should state clearly in the Discussion that yields are from small plots and – though the comparative effects across treatments might be acceptable – the actual yields would not be representative of those from full-sized fields.
RESPONSE 2: A sentence has been included in the discussion section: see lines 592-596 ‘However, it is important to note that the yield reported for the plot size used in this study (8 m x 1.2 m) may not fully represent actual yields in full-sized fields. While the comparative effects across treatments might be reliable, the actual yields observed in these plots are not necessarily indicative of those from larger, full-scale fields.’
- The new information on other met measurements is acceptable with the following exception – if photon flux is the primary measure of incoming solar, why are data in Table 2 presented in MJ, which implies total solar radiation?
REPONSE 3: This was an error and has been corrected in the Materials and Methods section; please see lines 210-211. ‘Solar radiation was measured with a pyranometer (model CM11, Kipp & Zonen Delft, Holland).’
IB, 1D, IE and IF. Amendments and responses acceptable.
Response 2. This is a plausible explanation.
Response 3. The explanation in the paragraph beginning “Secondly, the approach ..” does not stand scrutiny. There have been many previous analyses of the importance of weather in specific developmental phases to final outcomes. The authors should not claim that their approach here is radically new.
RESPONSE 4:
We agree with this and have highlighted the work of Dr. Fischer. A few sentences about this are shown in the discussion section in lines 682-690 “Together, these results may explain the higher yields observed in early va-rieties in this study. These findings align with the widely accepted concept that temperature and solar radiation are key weather factors influencing yield variability across wheat varieties. Dr. Fischer [47] emphasized this by demonstrating that yield potential fluctuations are strongly linked to the photothermal quotient (the ratio of solar radiation to mean temperature) in the month leading up to flowering, which affects kernel number, and to the temperature during grain filling, which plays a crucial role in determining kernel weight [47].”
The response to “Regarding the last question….” The arguments given do not cover all possibilities. Yes, there is a strong relation between degree days and maximum temperature, but this is to be expected since the maximum will be highly correlated with the mean. The question unresolved is why increasing in maximum temperature is negatively correlated with grain yield. The analysis so far has not distinguished the following possibilities:
- That yield is reduced by a direct effect of high temperature inhibiting grain filling or related processes – which is what the authors seem to prefer.
- That increasing temperature shortens the duration of grain filling (which is a commonly observed effect) and by doing so reduces the amount of solar radiation that the crops capture during this phase and therefore the amount of CO2 taken in and dry matter produced.
- That the genotypes have inherently different responses to high temperature.
- That other factors such as crop cover (affecting solar radiation interception) and capacity for translocation vegetative dry matter to grain also differ among varieties.
RESPONSE 5: All these reasons have been incorporated in the discussion section, see lines 597-607: ‘Elevated temperatures adversely affect grain yield through several interconnected mechanisms. High temperatures can directly inhibit grain filling and related processes, leading to reduced yields. Additionally, increased temperatures commonly shorten the grain filling period and accelerate crop senescence, reducing the time crops have to capture solar radiation, absorb CO2, and produce dry matter. The impact of elevated temperatures also varies among genotypes, as different varieties have distinct responses to heat stress. Furthermore, factors like crop cover, which affects how much solar radiation is intercepted, and the ability to translocate vegetative dry matter to grain, differ among varieties, further influencing yield outcomes under high-temperature conditions [5, 12-17].’
Data in the new Table 4 suggest yield is more strongly correlated with the duration of grain filling than with maximum temperature, which is consistent with the second option above – that high temperature is speeding up development and simply reducing the period over which the crop absorbs solar radiation. The paper would benefit from a discussion on these possibilities and a statement that a final resolution is not possible with the data gathered. The new section on threshold temperature in the first paragraph of the Discussion needs to be re-thought.
RESPONSE 6: Agreed. We have introduced this in the discussion section see lines 668-690 sentence explaining this: “Specifically, early varieties experienced lower potential evapotranspiration and temperatures during the vegetative stage; however, they were exposed to higher evapotranspiration and lower temperatures during the grain-filling stage. Additionally, early varieties exhibited an extended grain-filling period, averaging about 9 days longer than later varieties. The lower temperatures during grain filling (3.3 °C lower) and the longer grain-filling duration, combined with increased ETo during grain filling (ETo GF), likely contributed to greater total photosynthetic activity and increased availability of photo-assimilates for grain filling and yield formation. With this strategy, early varieties may keep photosynthesis active during the most crucial period of grain filling, maximizing yield formation. Despite the ex-tended grain filling duration, the total ETo from sowing to maturity in early varieties was lower than in late varieties, resulting in significant water sav-ings throughout the crop cycle—particularly advantageous in water-limited conditions (Table 5). Together, these results may explain the higher yields observed in early varieties in this study. These findings align with the widely accepted concept that temperature and solar radiation are key weather fac-tors influencing yield variability across wheat varieties. Dr. Fischer [47] emphasized this by demonstrating that yield potential fluctuations are strongly linked to the photothermal quotient (the ratio of solar radiation to mean temperature) in the month leading up to flowering, which affects kernel number, and to the temperature during grain filling, which plays a crucial role in determining kernel weight [47].”
Regarding the Reviewers request ‘The new section on threshold temperature in the first paragraph of the Discussion needs to be re-thought.’ We have removed the sentence regarding the phenology threshold due to the observed linear decrease in yield with phenology. Additional data, particularly from even earlier-maturing varieties, would be required to accurately determine the threshold. This would be an important and useful piece of information for which we will develop a new experimental setup.
Response 4. This change in placement greatly improves the flow of the Results. A very useful figure.
Response 5. A plausible explanation but the apparent discrepancy still needs explaining in full.
RESPONSE 7: The weather data presented in Table 2, with monthly averages, represent the means of daily records for each calendar month. These are calendar months and are independent of the crop's developmental stages. However, when considering the shorter grain-filling period observed in Sucs 2022, which was reduced by 10 days compared to Almacelles 2021, the cumulative potential evapotranspiration (the sum of daily ETo during grain filling) was lower. We have provided the calculations in the attached Excel file, containing the data used in the manuscript, for the reviewer's confirmation. Additionally, we have revised the legends for Figure 2 and Table 2 to clarify this further.
Response 6. An acceptable explanation but what are ‘units’ of nitrogen – N in soil or applied as fertilizer is usually quantified in kg/ha or similar? Please clarify. And while the reason for the high existing soil N is explained in the response, it needs explaining in the Methods, since it is very unusual for a high-N requiring crops such as wheat to have no N applied.
RESPONSE 8: These are Kg/Ha and are shown in lines 127-129. The high nitrogen content is primarily due to the large-scale pig production in the region and the frequent application of pig slurry to most fields throughout the year. Although policies are now being implemented to reduce excessive nitrogen accumulation from pig slurry discharges, the buildup over the years has been substantial. Many experimental stations are affected by this issue, and it is now more common to apply no additional nitrogen than to add it. This situation, while unfortunate and somewhat unique, has persisted for several years, but we hope to see improvements in the near future.
Response 7. Acceptable.
Further comments
RESPONSE 9: All amendments have been implemented.
Table 3. Please check all abbreviations and units e.g. degree days is usually given in units of oCd not oC.
Table 4 – difficult to tell which heading refers to a given column – e.g. is it TM V or VETo; no need to repeat as in ‘days to heading (days)’; define GDD and provide units.
Fig. 1. Giving dates on the horizontal axis as month/day/year every four days or so – and without a line drawn for the ais - makes it very difficult to work out the timings of the varieties. The axis should be simply numbered (e.g. by day or year) with tick marks every ten days or thereabout and additional pointers to indicate months.
Fig. 2 and elsewhere – units of mass – ton is not generally used in the SI systems, why not use t ha-1.

Reviewer 3 Report
Comments and Suggestions for Authors
Thank you for your work
Author Response
Thank you for the helpful comments.